# High-level cognition during story listening is reflected in high-order dynamic correlations in neural activity patterns

Lucy L. W. Owen [1], Thomas H. Chang[1,2] & Jeremy R. Manning [1✉]

Our thoughts arise from coordinated patterns of interactions between brain structures that change with our ongoing experiences. High-order dynamic correlations in neural activity patterns reflect different subgraphs of the brain's functional connectome that display homologous lower-level dynamic correlations. Here we test the hypothesis that high-level cognition is reflected in high-order dynamic correlations in brain activity patterns. We develop an approach to estimating high-order dynamic correlations in timeseries data, and we apply the approach to neuroimaging data collected as human participants either listen to a ten-minute story or listen to a temporally scrambled version of the story. We train across-participant pattern classifiers to decode (in held-out data) when in the session each neural activity snapshot was collected. We find that classifiers trained to decode from high-order dynamic correlations yield the best performance on data collected as participants listened to the (unscrambled) story. By contrast, classifiers trained to decode data from scrambled versions of the story yielded the best performance when they were trained using first-order dynamic correlations or non-correlational activity patterns. We suggest that as our thoughts become more complex, they are reflected in higher-order patterns of dynamic network interactions throughout the brain.

[1] Department of Psychological and Brain Sciences, Dartmouth College, Hanover, NH, USA. [2] Amazon.com, Seattle, WA, USA.
✉email: jeremy.r.manning@dartmouth.edu

A central goal in cognitive neuroscience is to elucidate the neural code: i.e., the mapping between (a) mental states or cognitive representations and (b) neural activity patterns. One means of testing models of the neural code is to ask how accurately that model is able to "translate" neural activity patterns into known (or hypothesized) mental states or cognitive representations[1–9]. Training decoding models on different types of neural features (Fig. 1a) can also help to elucidate which specific aspects of neural activity patterns are informative about cognition and, by extension, which types of neural activity patterns might compose the neural code. For example, prior work has used region of interest analyses to estimate the anatomical locations of specific neural representations[10], or to compare the relative contributions to the neural code of multivariate activity patterns versus dynamic correlations between neural activity patterns[11,12]. An emerging theme in this literature is that cognition is mediated by dynamic interactions between brain structures[13–25].

Studies of the neural code to date have primarily focused on univariate or multivariate neural patterns[2], or (more recently) on patterns of dynamic first-order correlations (i.e., interactions between pairs of brain structures[11,12,18,20–22]). What might the future of this line of work hold? For example, is the neural code implemented through higher-order interactions between brain structures[26]? Second-order correlations reflect homologous patterns of correlation. In other words, if the dynamic patterns of correlations between two regions, A and B, are similar to those between two other regions, C and D, this would be reflected in the second-order correlations between (A–B) and (C–D). In this way, second-order correlations identify similarities and differences between subgraphs of the brain's connectome. Analogously, third-order correlations reflect homologies between second-order correlations– i.e., homologous patterns of homologous interactions between brain regions. More generally, higher-order correlations reflect homologies between patterns of lower-order correlations. We can then ask: which "orders" of interaction are most reflective of high-level cognitive processes?

One reason one might expect to see homologous networks in a dataset is related to the notion that network dynamics reflect ongoing neural computations or cognitive processing[27]. If the nodes in two brain networks are interacting (within each network) in similar ways then, according to our characterization of network dynamics, we refer to the similarities between those patterns of interaction as higher-order correlations. When higher-order correlations are themselves changing over time, we can also attempt to capture and characterize those high-order dynamics.

Another central question pertains to the extent to which the neural code is carried by activity patterns that directly reflect ongoing cognition[1,2], versus the dynamic properties of the network structure itself, independent of specific activity patterns in any given set of regions[16]. For example, graph measures such as centrality and degree[28] may be used to estimate how a given brain structure is "communicating" with other structures, independently of the specific neural representations carried by those structures. If one considers a brain region's position in the network (e.g., its eigenvector centrality) as a dynamic property, one can compare how the positions of different regions are correlated, and/or how those patterns of correlations change over time. We can also compute higher-order patterns in these correlations to characterize homologous subgraphs in the connectome that

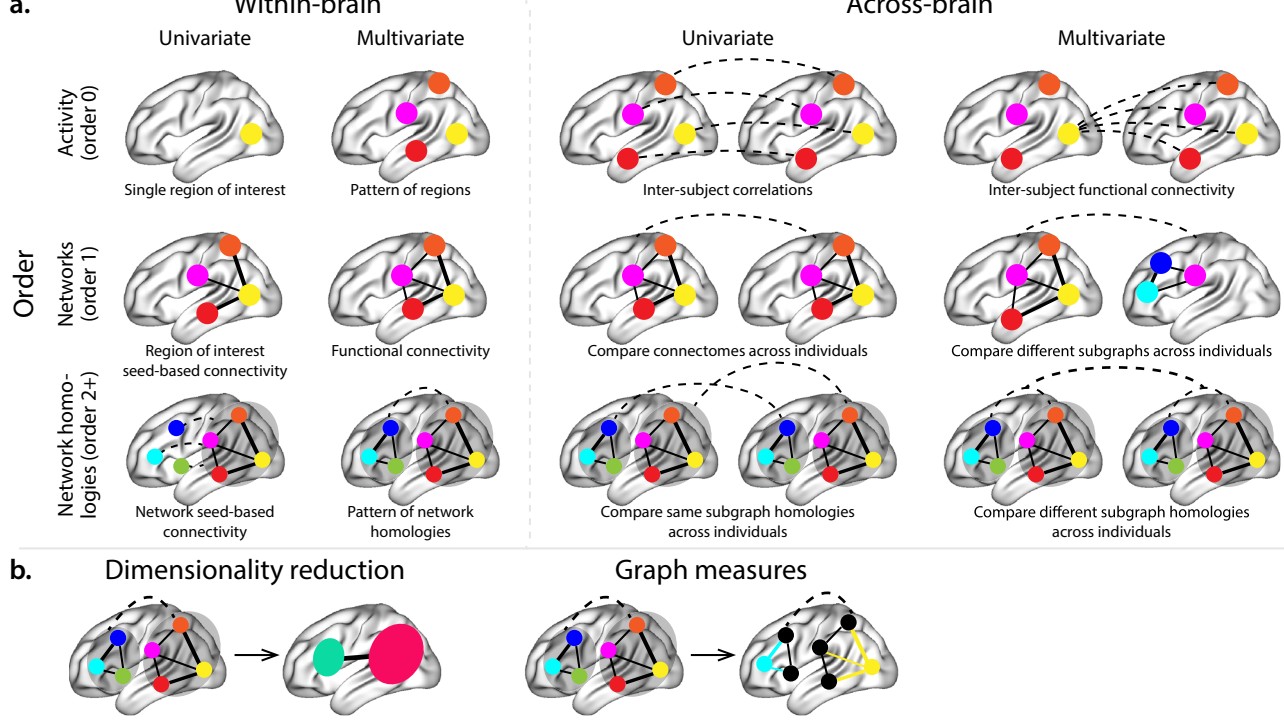

**Fig. 1 Neural patterns. a** A space of neural features. Within-brain analyses are carried out within a single brain, whereas across-brain analyses compare neural patterns across two or more individuals' brains. Univariate analyses characterize the activities of individual units (e.g., nodes, small networks, hierarchies of networks, etc.), whereas multivariate analyses characterize the patterns of activity across units. Order 0 patterns involve individual nodes; order 1 patterns involve node-node interactions; order 2 (and higher) patterns relate to interactions between homologous networks. Each of these patterns may be static (e.g., averaging over time) or dynamic. **b** Summarizing neural patterns. To efficiently compute with complex neural patterns, it can be useful to characterize the patterns using summary measures. Dimensionality reduction algorithms project the patterns onto lower-dimensional spaces whose dimensions reflect weighted combinations or nonlinear transformations of the dimensions in the original space. Graph measures characterize each unit's participation in its associated network.

display similar changes in their constituent brain structures' interactions with the rest of the brain.

To gain insights into the above aspects of the neural code, we developed a computational framework for estimating dynamic high-order correlations in timeseries data. This framework provides an important advance, in that it enables us to examine patterns of higher-order correlations that are computationally intractable to estimate via conventional methods. Given a multivariate timeseries, our framework provides timepoint-by-timepoint estimates of the first-order correlations, second-order correlations, and so on. Our approach combines a kernel-based method for computing dynamic correlations in timeseries data with a dimensionality reduction step (Fig. 1b) that projects the resulting dynamic correlations into a low-dimensional space. We explored two dimensionality reduction approaches: principle components analysis[29] (PCA), which preserves an approximately invertible transformation back to the original data[30–32], and a second non-invertible algorithm for computing dynamic patterns in eigenvector centrality[33]. This latter approach characterizes correlations between each feature dimension's relative position in the network (at each moment in time) in favor of the specific activity histories of different features[26,34,35].

We validated our approach using synthetic data where the underlying correlations were known. We then applied our framework to a neuroimaging dataset collected as participants listened to either an audio recording of a ten-minute story, listened to a temporally scrambled version of the story, or underwent a resting state scan[36]. Temporal scrambling has been used in a growing number of studies, largely by Uri Hasson's group, to identify brain regions that are sensitive to higher-order and longer-timescale information (e.g., cross-sensory integration, rich narrative meaning, complex situations, etc.) versus regions that are primarily sensitive to low-order (e.g., sensory) information. For example,[37] argues that when brain areas are sensitive to fine versus coarse temporal scrambling, this indicates that they are "higher order" in the sense that they process contextual information pertaining to further-away timepoints. By contrast, low-level regions, such as primary sensory cortices, do not meaningfully change their responses (after correcting for presentation order) even when the stimulus is scrambled at fine timescales.

We used a subset of the story listening and rest data to train across-participant classifiers to decode listening times (of groups of participants) using a blend of neural features (comprising neural activity patterns, as well as different orders of dynamic correlations between those patterns that were inferred using our computational framework). We found that both the PCA-based and eigenvector centrality-based approaches yielded neural patterns that could be used to decode accurately (i.e., well above chance). Both approaches also yielded the best decoding accuracy for data collected during (intact) story listening when high-order (PCA: second-order; eigenvector centrality: fourth-order) dynamic correlation patterns were included as features. When we trained classifiers on the scrambled stories or resting state data, only (relatively) lower-order dynamic patterns were informative to the decoders. Taken together, our results indicate that high-level cognition is supported by high-order dynamic patterns of communication between brain structures.

## Results

We sought to understand whether high-level cognition is reflected in dynamic patterns of high-order correlations. To that end, we developed a computational framework for estimating the dynamics of stimulus-driven high-order correlations in multivariate timeseries data (see Dynamic inter-subject functional connectivity (DISFC) and Dynamic higher-order correlations).

We evaluated the efficacy of this framework at recovering known patterns in several synthetic datasets (see Synthetic data: simulating dynamic first-order correlations and Synthetic data: simulating dynamic higher-order correlations). We then applied the framework to a public fMRI dataset collected as participants listened to an auditorily presented story, listened to a temporally scrambled version of the story, or underwent a resting state scan (see Functional neuroimaging data collected during story listening). We used the relative decoding accuracies of classifiers trained on different sets of neural features to estimate which types of features reflected ongoing cognitive processing.

**Recovering known dynamic first-order correlations**. We generated synthetic datasets that differed in how the underlying first-order correlations changed over time. For each dataset, we applied Eq. (4) with a variety of kernel shapes and widths. We assessed how well the true underlying correlations at each timepoint matched the recovered correlations (Fig. 2). For every kernel and dataset we tested, our approach recovered the correlation dynamics we embedded into the data. However, the quality of these recoveries varied across different synthetic datasets in a kernel-dependent way.

In general, wide monotonic kernel shapes (Laplace, Gaussian), and wider kernels (within a shape), performed best when the correlations varied gradually from moment-to-moment (Fig. 2a, c, d). In the extreme, as the rate of change in correlations approaches 0 (Fig. 2a), an infinitely wide kernel would exactly recover the Pearson's correlation (e.g., compare Eqs. (1) and (4)).

When the correlation dynamics were unstructured in time (Fig. 2b), a Dirac $\delta$ kernel (infinitely narrow) performed best. This is because, when every timepoint's correlations are independent of the correlations at every other timepoint, averaging data over time dilutes the available signal. Following a similar pattern, holding kernel shape fixed, narrower kernel parameters better recovered randomly varying correlations.

**Recovering known dynamic higher-order correlations**. Following our approach to evaluating our ability to recover known dynamic first-order correlations from synthetic data, we generated an analogous second set of synthetic datasets that we designed to exhibit known dynamic first-order and second-order correlations (see Synthetic data: simulating dynamic higher-order correlations). We generated a total of 400 datasets (100 datasets for each category) that varied in how the first-order and second-order correlations changed over time. We then repeatedly applied Eq. (4) using the overall best-performing kernel from our first-order tests (a Laplace kernel with a width of 20; Fig. 2) to assess how closely the recovered dynamic correlations matched the dynamic correlations we had embedded into the datasets.

Overall, we found that we could reliably recover both first-order and second-order correlations from the synthetic data (Fig. 3). When the correlations were stable for longer intervals, or changed gradually (constant, ramping, and event datasets), recovery performance was relatively high, and we were better able to recover dynamic first-order correlations than second-order correlations. This is because errors in our estimation procedure at lower orders necessarily propagate to higher orders (since lower-order correlations are used to estimate higher-order correlations). Conversely, when the correlations were particularly unstable (random datasets), we better recovered second-order correlations. This is because noise in our data generation procedure propagates from higher orders to lower orders (see Synthetic data: simulating dynamic high-order correlations).

We also examined the impact of the data duration (Fig. S3) and complexity (number of zero-order features; Fig. S4) on our ability

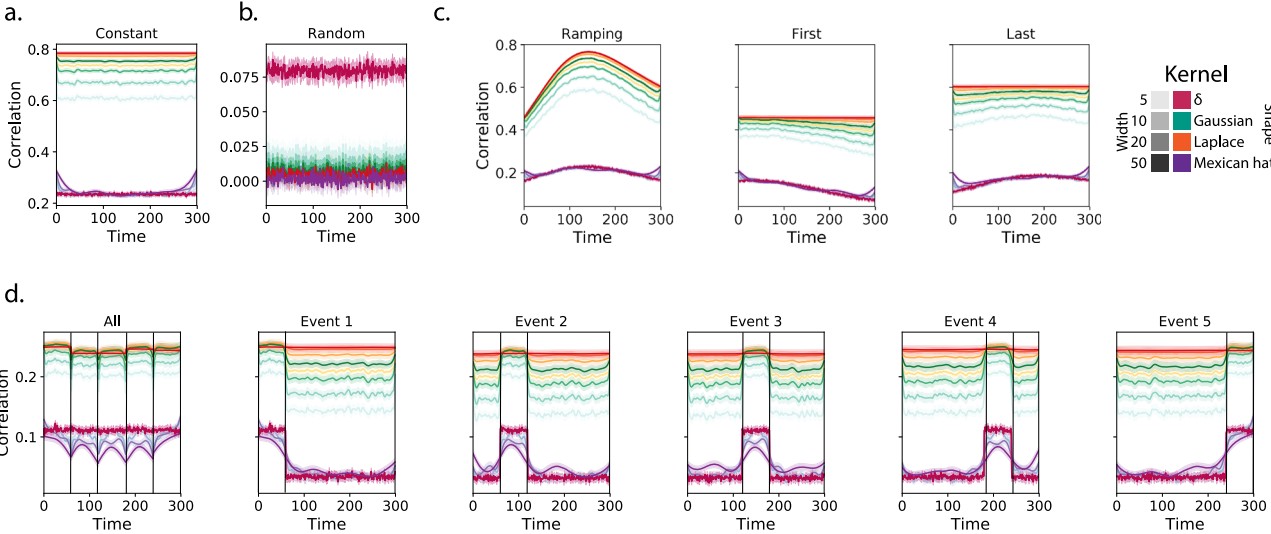

**Fig. 2 Recovering known dynamic first-order correlations from synthetic data.** Each panel displays the average correlations between the vectorized upper triangles of the recovered correlation matrix at each timepoint and either the true underlying correlation at each timepoint or a reference correlation matrix (the averages are taken across 100 different randomly generated synthetic datasets of each given category, each with $K = 50$ features and $T = 300$ timepoints). Error ribbons denote 95% confidence intervals of the mean (taken across datasets). Different colors denote different kernel shapes, and the shading within each color family denotes the kernel width parameter. For a complete description of each synthetic dataset, see Synthetic data: simulating dynamic first-order correlations. **a** Constant correlations. These datasets have a stable (unchanging) underlying correlation matrix. **b** Random correlations. These datasets are generated using a new independently drawn correlation matrix at each new timepoint. **c** Ramping correlations. These datasets are generated by smoothly varying the underlying correlations between the randomly drawn correlation matrices at the first and last timepoints. The left panel displays the correlations between the recovered dynamic correlations and the underlying ground truth correlations. The middle panel compares the recovered correlations with the first timepoint's correlation matrix. The right panel compares the recovered correlations with the last timepoint's correlation matrix. **d** Event-based correlations. These datasets are each generated using five randomly drawn correlation matrices that each remain stable for a fifth of the total timecourse. The left panel displays the correlations between the recovered dynamic correlations and the underlying ground truth correlations. The right panels compare the recovered correlations with the correlation matrices unique to each event. The vertical lines denote event boundaries. Source data are provided as a Source Data file.

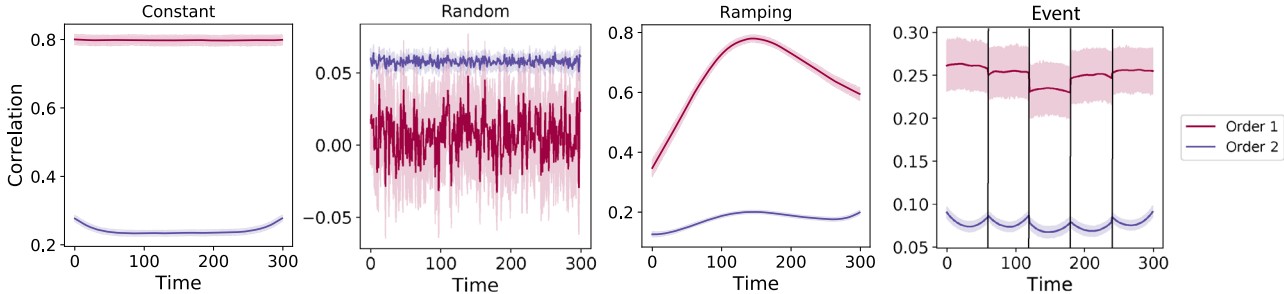

**Fig. 3 Recovery of simulated first-order and second-order dynamic correlations.** Each panel displays the average correlations between the vectorized upper triangles of the recovered first-order and second-order correlation matrices and the true (simulated) first-order and second-order correlation matrices at each timepoint and for each synthetic dataset. (The averages are taken across 100 different randomly generated synthetic datasets of each given category, each with $K = 10$ features and $T = 300$ timepoints.) Error ribbons denote 95% confidence intervals of the mean (taken across datasets). For a complete description of each synthetic dataset, see synthetic data: simulating dynamic higher-order correlations. All estimates represented in this figure were computed using a Laplace kernel (width = 20). Constant. These datasets have stable (unchanging) underlying second-order correlation matrices. Random. These datasets are generated using a new independently drawn second-order correlation matrix at each timepoint. Ramping. These datasets are generated by smoothly varying the underlying second-order correlations between the randomly drawn correlation matrices at the first and last timepoints. Event. These datasets are each generated using five randomly drawn second-order correlation matrices that each remain stable for a fifth of the total timecourse. The vertical lines denote event boundaries. Note that the "dips" and "ramps" at the boundaries of sharp transitions (e.g., the beginning and ends of the "constant" and "ramping" datasets, and at the event boundaries of the "event" datasets) are finite-sample effects that reflect the reduced numbers of samples that may be used to accurately estimate correlations at sharp boundaries. Source data are provided as a Source Data file.

to accurately recover ground truth first-order and second-order dynamic correlations. In general, we found that our approach better recovers ground truth dynamic correlations from longer duration timeseries data. We also found that our approach tends to best recover data generated using fewer zero-order features (i.e., lower complexity), although this tendency was not strictly

monotonic. Further, because our data generation procedure requires $\mathcal{O}(K^4)$ memory to generate a second-order timeseries with $K$ zero-order features, we were not able to fully explore how the number of zero-order features affects recovery accuracy as the number of features gets larger (e.g., as it approaches the number of features present in the fMRI data we examine below). Although

we were not able to formally test this to our satisfaction, we expect that accurately estimating dynamic high-order correlations would require data with many more zero-order features than we were able to simulate. Our reasoning is that high-order correlations necessarily involve larger numbers of lower-order features, so achieving adequate "resolution" high-order timeseries might require many low-order features.

Taken together, our explorations using synthetic data indicated that we are able to partially, but not perfectly, recover ground truth dynamic first-order and second-order correlations. This suggests that our modeling approach provides a meaningful (if noisy) estimate of high-order correlations. We next turned to analyses of human fMRI data to examine whether the recovered dynamics might reflect the dynamics of human cognition during a naturalistic story-listening task.

**Cognitively relevant dynamic high-order correlations in fMRI data.** We used across-participant temporal decoders to identify cognitively relevant neural patterns in fMRI data (see Forward inference and decoding accuracy). The dataset we examined[36] comprised four experimental conditions that exposed participants to stimuli that varied systematically in how cognitively engaging they were. The intact experimental condition (intact) had participants listen to an audio recording of a 10-min story. The paragraph-scrambled experimental condition (paragraph) had participants listen to a temporally scrambled version of the story, where the paragraphs occurred out of order (but where the same total set of paragraphs were presented over the full listening interval). All participants in this condition experienced the scrambled paragraphs in the same order. The word-scrambled experimental condition (word) had participants listen to a temporally scrambled version of the story where the words in the story occurred in a random order. All participants in the word condition experienced the scrambled words in the same order. Finally, in a rest experimental condition (rest), participants lay in the scanner with no overt stimulus, with their eyes open (blinking as needed). This public dataset provided a convenient means of testing our hypothesis that different levels of cognitive processing and engagement are reflected in different orders of brain activity dynamics.

In brief, we computed timeseries of dynamic high-order correlations that were similar across participants in each of two randomly assigned groups: a training group and a test group. We then trained classifiers on the training group's data to match each sample from the test group with a stimulus timepoint. Each classifier comprised a weighted blend of neural patterns that reflected up to $n$th-order dynamic correlations (see Feature weighting and testing). We repeated this process for $n \in \{0, 1, 2, ..., 10\}$. Our examinations of synthetic data suggested that none of the kernels we examined were "universal" in the sense of optimally recovering underlying correlations regardless of the temporal structure of those correlations. We found a similar pattern in the (real) fMRI data, whereby different kernels yielded different decoding accuracies, but no single kernel emerged as the clear "best." In our analyses of neural data, we therefore averaged our decoding results over a variety of kernel shapes and widths in order to identify results that were robust to specific kernel parameters (see Identifying robust decoding results).

Our approach to estimating dynamic high-order correlations entails mapping the high-dimensional feature space of correlations (represented by a $T$ by $\mathcal{O}(K^2)$ matrix) onto a lower-dimensional feature space (represented by a $T$ by $K$ matrix). We carried out two sets of analyses that differed in how this mapping was computed. The first set of analyses used PCA to find a low-

dimensional embedding of the original dynamic correlation matrices (Fig. 4a, b). The second set of analyses characterized correlations in dynamics of each feature's eigenvector centrality, but did not preserve the underlying activity dynamics (Fig. 4c, d).

Both sets of temporal decoding analyses yielded qualitatively similar results for the auditory (non-rest) conditions of the experiment (Fig. 4: pink, green, and teal lines; Fig. 5: three leftmost columns). The highest decoding accuracy for participants who listened to the intact (unscrambled) story was achieved using high-order dynamic correlations (PCA: second-order; eigenvector centrality: fourth-order). Scrambled versions of the story were best decoded by lower-order correlations (PCA/paragraph: first-order; PCA/word: order zero; eigenvector centrality/paragraph: order zero; and eigenvector centrality/word: order zero). The two sets of analyses yielded different decoding results on resting state data (Fig. 4: purple lines; Fig. 5: rightmost column). We note that, while the resting state times could be decoded reliably, the accuracies were only very slightly above chance. We speculate that the decoders might have picked up on attentional drift, boredom, or tiredness; we hypothesize that these all increased throughout the resting state scan. The decoders might be picking up on aspects of these loosely defined cognitive states that are common across individuals. The PCA-based approach achieved the highest resting state decoding accuracy using order zero features (non-correlational, activation-based), whereas the eigenvector centrality-based approach achieved the highest resting state decoding accuracy using second-order correlations. Taken together, these analyses indicate that high-level cognitive processing (while listening to the intact story) is reflected in the dynamics of high-order correlations in brain activity, whereas lower-level cognitive processing (while listening to scrambled versions of the story that lack rich meaning) is reflected in the dynamics of lower-order correlations and non-correlational activity dynamics. Further, these patterns are associated both with the underlying activity patterns (characterized using PCA) and also with the changing relative positions that different brain areas occupy in their associated networks (characterized using eigenvector centrality).

Having established that patterns of high-order correlations are informative to decoders, we next wondered which specific networks of brain regions contributed most to these patterns. As a representative example, we selected the kernel parameters that yielded decoding accuracies that were the most strongly correlated (across conditions and orders) with the average accuracies across all of the kernel parameters we examined. Using Fig. 4c as a template, the best-matching kernel was a Laplace kernel with a width of 50 (see Kernel-based approach for computing dynamic correlations and Fig. S9). We used this kernel to compute a single $K$ by $Kn$th-order DISFC matrix for each experimental condition. We then used Neurosynth[38] to compute the terms most highly associated with the most strongly correlated pairs of regions in each of these matrices (Fig. 6; see Reverse inference).

For all of the story listening conditions (intact, paragraph, and word; top three rows of Fig. 6), we found that first- and second-order correlations were most strongly associated with auditory and speech processing areas. During intact story listening, third-order correlations reflected integration with visual areas, and fourth-order correlations reflected integration with areas associated with high-level cognition and cognitive control, such as the ventrolateral prefrontal cortex. However, when participants listened to temporally scrambled stories, these higher-order correlations instead involved interactions with additional regions associated with speech and semantic processing (second and third rows of Fig. 6). By contrast, we found a much different set of patterns in the resting state data (Fig. 6, bottom row). First-order

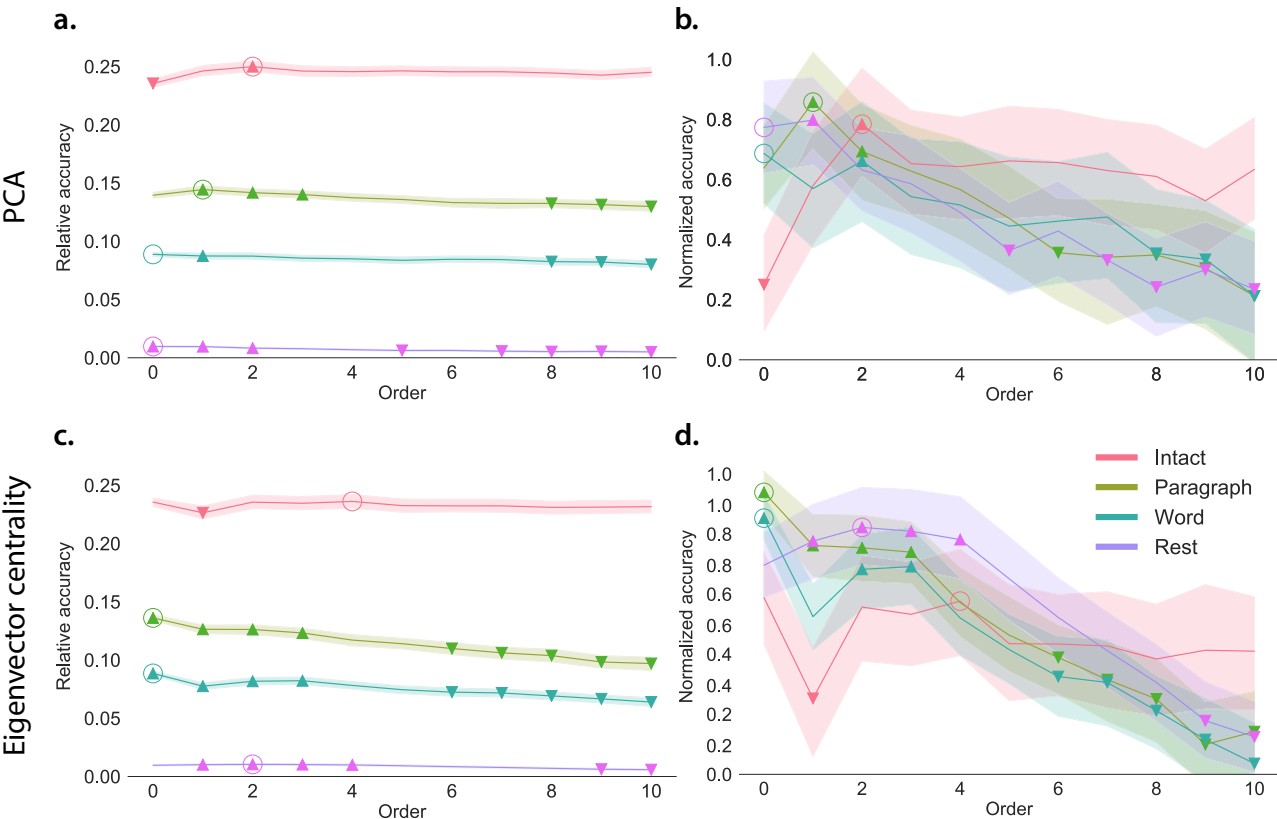

**Fig. 4 Across-participant timepoint decoding accuracy varies with correlation order and cognitive engagement. a** Decoding accuracy as a function of order: PCA. "Order'' (*x*-axis) refers to the maximum order of dynamic correlations that were available to the classifiers (see Feature weighting and testing). The reported across-participant decoding accuracies are averaged over all kernel shapes and widths (see Identifying robust decoding results). The *y*-values are displayed relative to chance accuracy (intact: $\frac{1}{300}$; paragraph: $\frac{1}{272}$; word: $\frac{1}{300}$; rest: $\frac{1}{400}$; these chance accuracies were subtracted from the observed accuracies to obtain the relative accuracies reported on the *y*-axis). The error ribbons denote 95% confidence intervals of the means across cross-validation folds (i.e., random assignments of participants to the training and test sets). The colors denote the experimental condition. Arrows denote sets of features that yielded reliably higher (upward facing) or lower (downward facing) decoding accuracy than the mean of all other features (via a two-tailed *t*-test, thresholded at *p* < 0.05). Figure 5 displays additional comparisons between the decoding accuracies achieved using different sets of neural features. The circled values represent the maximum decoding accuracy within each experimental condition. **b** Normalized timepoint decoding accuracy as a function of order: PCA. This panel displays the same results as Panel **a**, but here each curve has been normalized to have a maximum value of 1 and a minimum value of 0 (including the upper and lower bounds of the respective 95% confidence intervals of the mean). Panels **a** and **b** used PCA to project each high-dimensional pattern of dynamic correlations onto a lower-dimensional space. **c** Timepoint decoding accuracy as a function of order: eigenvector centrality. This panel is in the same format as Panel **a**, but here eigenvector centrality has been used to project the high-dimensional patterns of dynamic correlations onto a lower-dimensional space. **d** Normalized timepoint decoding accuracy as a function of order: eigenvector centrality. This panel is in the same format as Panel **b**, but here eigenvector centrality has been used to project the high-dimensional patterns of dynamic correlations onto a lower-dimensional space. See Figs. S1 and S2 for decoding results broken down by kernel shape and width, respectively. Source data are provided as a Source Data file.

resting state correlations were most strongly associated with regions involved in counting and numerical understanding. Second-order resting state correlations were strongest in visual areas; third-order correlations were strongest in task-positive areas; and fourth-order correlations were strongest in regions associated with autobiographical and episodic memory. We carried out analogous analyses to create maps (and decode the top associated Neurosynth terms) for up to 15th-order correlations (Figs. S5–S8). Of note, examining 15th-order correlations between 700 nodes using conventional methods would have required storing roughly $\frac{700^{2 \times 15}}{2} \approx 1.13 \times 10^{85}$ floating point numbers– assuming single-precision (32 bits each), this would require roughly 32 times as many bits as there are molecules in the known universe! Although these 15th-order correlations do appear (visually) to have some well-formed structure, we provide this latter example primarily as a demonstration of the efficiency and scalability of our approach.

## Discussion

We tested the hypothesis that high-level cognition is reflected in high-order brain network dynamics[19,26]. We examined high-order network dynamics in functional neuroimaging data collected during a story listening experiment. When participants listened to an auditory recording of the story, participants exhibited similar high-order brain network dynamics. By contrast, when participants instead listened to temporally scrambled recordings of the story, only lower-order brain network dynamics were similar across participants. Our results indicate that higher orders of network interactions support higher-level aspects of cognitive processing (Fig. 7).

The notion that cognition is reflected in (and possibly mediated by) patterns of first-order network dynamics has been suggested by or proposed in myriad empirical studies and reviews[11,12,17,18,20–22,24,25,32,39–42]. Our study extends this line of work by finding cognitively relevant higher-order network

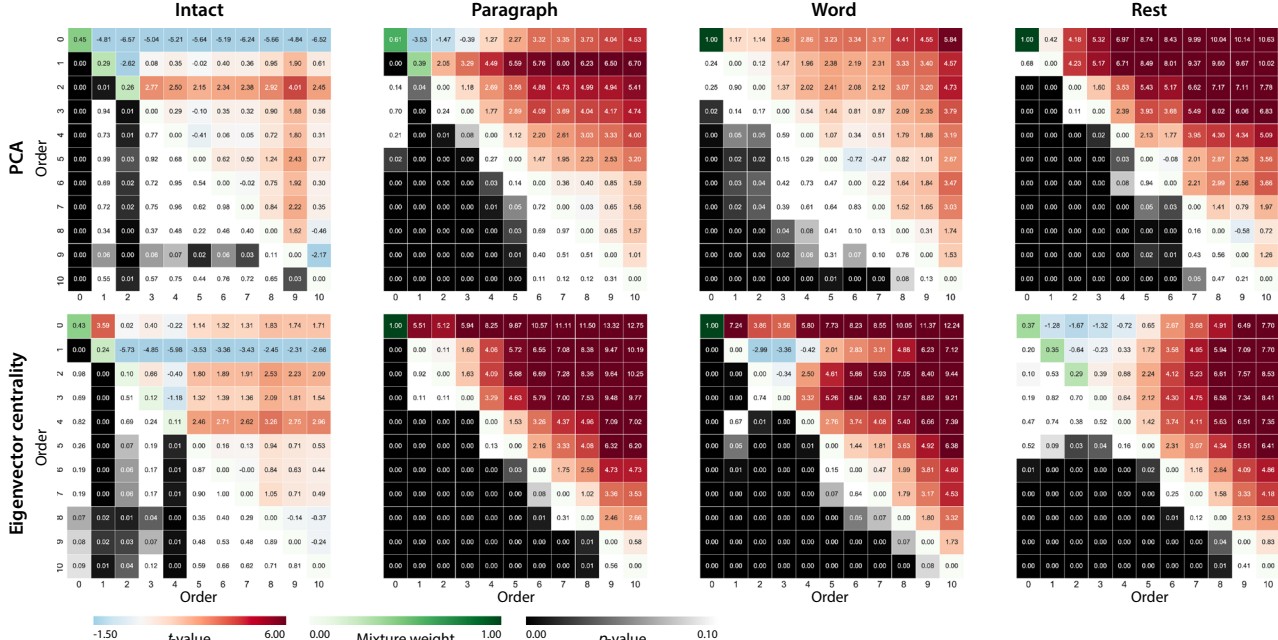

**Fig. 5 Statistical summary of decoding accuracies for different neural features.** Each column of matrices displays decoding results for one experimental condition (intact, paragraph, word, and rest). We considered dynamic activity patterns (order 0) and dynamic correlations at different orders (order > 0). We used two-tailed *t*-tests to compare the distributions of decoding accuracies obtained using each pair of features. The distributions for each feature reflect the set of average decoding accuracies (across all kernel parameters), obtained for each random assignment of training and test groups. In the upper triangles of each matrix, warmer colors (positive *t*-values) indicate that the neural feature indicated in the given row yielded higher accuracy than the feature indicated in the given column. Cooler colors (negative *t*-values) indicate that the feature in the given row yielded lower decoding accuracy than the feature in the given column. The lower triangles of each map denote the corresponding *p*-values for the *t*-tests. The diagonal entries display the relative average optimized weight given to each type of feature in a decoder that included all feature types (see Feature weighting and testing). Source data are provided as a Source Data file.

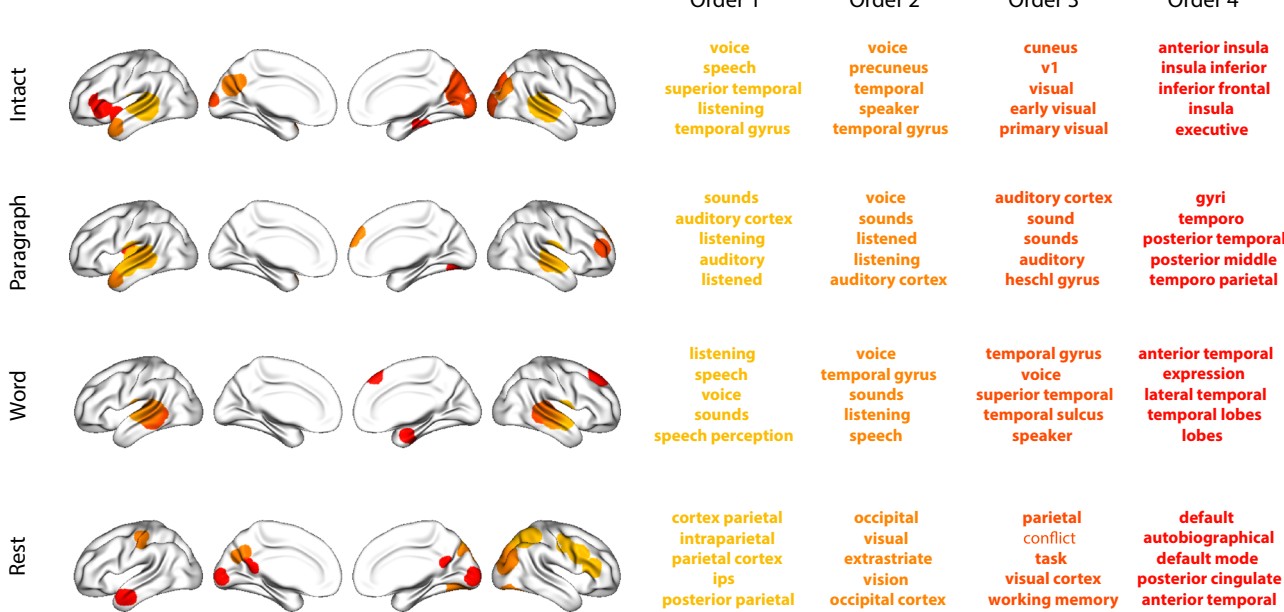

**Fig. 6 Top terms associated with the most strongly correlated nodes at each order.** Each color corresponds to one order of inter-subject functional correlations. To calculate the dynamic correlations, eigenvector centrality has been used to project the high-dimensional patterns of dynamic correlations onto a lower-dimensional space at each previous order, which allows us to map the brain regions at each order by retaining the features of the original space. The inflated brain plots display the locations of the endpoints of the 10 strongest (absolute value) correlations at each order, thresholded at 0.999, and projected onto the cortical surface[91]. The lists of terms on the right display the top five Neurosynth terms[38] decoded from the corresponding brain maps for each order. Each row displays data from a different experimental condition. Additional maps and their corresponding Neurosynth terms may be found in the Supplementary materials (intact: Fig. S5; paragraph: Fig. S6; word: Fig. S7; rest: Fig. S8). Source data are provided as a Source Data file.

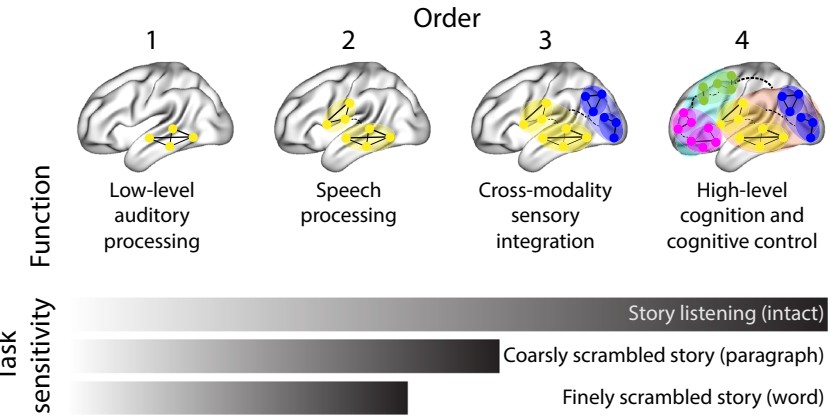

**Fig. 7 Proposed high-order network dynamics underlying high-level cognition during story listening.** Schematic depicts higher orders of network interactions supporting higher-level aspects of cognitive processing. When tasks evoke richer, deeper, and/or higher-level processing, this is reflected in higher-order network interactions.

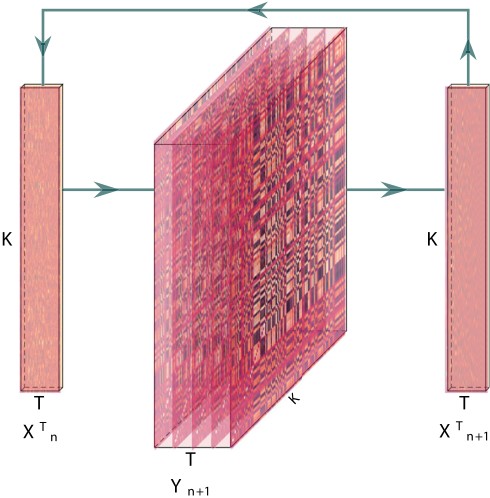

**Fig. 8 Estimating dynamic high-order correlations.** Given a $T$ by $K$ matrix of multivariate timeseries data, $\mathbf{X}_n$ (where $n \in \mathbb{N}, n \geq 0$), we use Eq. (4) to compute a timeseries of $K$ by $K$ correlation matrices, $\mathbf{Y}_{n+1}$. We then approximate $\mathbf{Y}_{n+1}$ with the $T$ by $K$ matrix, $\mathbf{X}_{n+1}$. This process may be repeated to scalably estimate iteratively higher-order correlations in the data. Note that the transposes of $\mathbf{X}_n$ and $\mathbf{X}_{n+1}$ are displayed in the figure for compactness.

dynamics that reflect ongoing cognition. Our findings also complement other work that uses graph theory and topology to characterize how brain networks reconfigure during cognition[16,26,30,31,34,35,43].

An open question not addressed by our study pertains to how different structures integrate incoming information with different time constants. For example, one line of work suggests that the cortical surface comprises a structured map such that nearby brain structures process incoming information at similar timescales. Low-level sensory areas integrate information relatively quickly, whereas higher-level regions integrate information relatively slowly[37,44–49]. A similar hierarchy appears to play a role in predicting future events[50]. Other related work in human and mouse brains indicates that the temporal response profile of a given brain structure may relate to how strongly connected that structure is with other brain areas[51]. Further study is needed to understand the role of temporal integration at different scales of network interaction, and across different anatomical structures. Importantly, our analyses do not speak to the physiological basis of higher-order dynamics, and could reflect nonlinearities, chaotic patterns, non-stationarities, and/

or multistability, etc. However, our decoding analyses do indicate that higher-order dynamics are consistent across individuals, and therefore unlikely to reflect non-stimulus-driven dynamics that are unlikely to be similar across individuals.

One limitation of our approach relates to how noise propagates in our estimation procedure. Specifically, our procedure for estimating high-order dynamic correlations depends on estimates of lower-order dynamic correlations. This means that our measures of which higher-order patterns are reliable and stable across experimental conditions are partially confounded with the stability of lower-order patterns. Prior work suggests that the stability of what we refer to here as first-order dynamics likely varies across the experimental conditions we examined[36]. Therefore a caveat to our claim that richer stimuli evoke more stable higher-order dynamics is that our approach assumes that those high-order dynamics reflect relations or interactions between lower-order features.

Another potential limitation of our approach relates to recent work suggesting that the brain undergoes rapid state changes, for example across event boundaries[44,52]. used hidden semi-Markov models to estimate state-specific network dynamics[53]. Our general approach might be extended by considering putative state transitions. For example, rather than weighting all timepoints using a similar kernel (Eq. (4)), the kernel function could adapt on a timepoint-by-timepoint basis such that only timepoints determined to be in the same "state" were given non-zero weight.

Identifying high-order network dynamics associated with high-level cognition required several important methods advances. First, we used kernel-based dynamic correlations to extend the notion of (static) inter-subject functional connectivity[36] to a DISFC that does not rely on sliding windows[11], and that may be computed at individual timepoints. This allowed us to precisely characterize stimulus-evoked network dynamics that were similar across individuals. Second, we developed a computational framework for efficiently and scalably estimating high-order dynamic correlations. Our approach uses dimensionality reduction algorithms and graph measures to obtain low-dimensional embeddings of patterns of network dynamics. Third, we developed an analysis framework for identifying robust decoding results by carrying out our analyses using a range of parameter values and identifying which results were robust to specific parameter choices. By showing that high-level cognition is reflected in high-order network dynamics, we have elucidated the next step on the path towards understanding the neural basis of cognition.

## Methods
Our general approach to efficiently estimating high-order dynamic correlations comprises four general steps (Fig. 8). First, we derive a kernel-based approach to computing

**Fig. 9 Examples of kernel functions.** Each panel displays per-timepoint weights for a kernel centered at $t = 50$, evaluated at 100 timepoints ($\tau \in [1, ..., 100]$). **a** Uniform kernel. The weights are timepoint-invariant; observations at all timepoints are weighted equally, and do not change as a function of $\tau$. This is a special case kernel function that reduces dynamic correlations to static correlations. **b** Dirac $\delta$ kernel. Only the observation at timepoint $t$ is given a non-zero weight (of 1). **c** Gaussian kernels. Each kernel's weights fall off in time according to a Gaussian probability density function centered on time $t$. Weights derived using several different example width parameters ($\sigma^2$) are displayed. **d** Laplace kernels. Each kernel's weights fall off in time according to a Laplace probability density function centered on time $t$. Weights derived using several different example width parameters ($b$) are displayed. **e** Mexican hat (Ricker wavelet) kernels. Each kernel's weights fall off in time according to a Ricker wavelet centered on time $t$. This function highlights the contrasts between local versus surrounding activity patterns in estimating dynamic correlations. Weights derived using several different example width parameters ($\sigma$) are displayed.

dynamic pairwise correlations in a $T$ (timepoints) by $K$ (features) multivariate timeseries, $\mathbf{X}_0$. This yields a $T$ by $\mathcal{O}(K^2)$ matrix of dynamic correlations, $\mathbf{Y}_1$, where each row comprises the upper triangle and diagonal of the correlation matrix at a single timepoint, reshaped into a row vector (this reshaped vector is $\left(\frac{K^2 - K}{2} + K\right)$-dimensional). Second, we apply a dimensionality reduction step to project the matrix of dynamic correlations back onto a $K$-dimensional space. This yields a $T$ by $K$ matrix, $\mathbf{X}_1$, that reflects an approximation of the dynamic correlations reflected in the original data. Third, we use repeated applications of the kernel-based dynamic correlation step to $\mathbf{X}_n$ and the dimensionality reduction step to the resulting $\mathbf{Y}_{n+1}$ to estimate high-order dynamic correlations. Each application of these steps to a $T$ by $K$ timeseries $\mathbf{X}_n$ yields a $T$ by $K$ matrix, $\mathbf{X}_{n+1}$, that reflects the dynamic correlations between the columns of $\mathbf{X}_n$. In this way, we refer to $n$ as the order of the timeseries, where $\mathbf{X}_0$ (order 0) denotes the original data and $\mathbf{X}_n$ denotes (approximated) $n$th-order dynamic correlations between the columns of $\mathbf{X}_0$. Finally, we use a cross-validation-based decoding approach to evaluate how well information contained in a given order (or weighted mixture of orders) may be used to decode relevant cognitive states. If including a given $\mathbf{X}_n$ in the feature set yields higher classification accuracy on held-out data, we interpret this as evidence that the given cognitive states are reflected in patterns of $n$th-order correlations.

All of the code used to produce the figures and results in this manuscript, along with links to the corresponding datasets, may be found at github.com/ContextLab/timecorr-paper. In addition, we have released a Python toolbox for computing dynamic high-order correlations in timeseries data; our toolbox may be found at timecorr.readthedocs.io.

**Kernel-based approach for computing dynamic correlations.** Given a $T$ by $K$ matrix of observations, $\mathbf{X}$, we can compute the (static) Pearson's correlation between any pair of columns, $\mathbf{X}(\cdot, i)$ and $\mathbf{X}(\cdot, j)$ using[29]:

$$\mathrm{corr}(\mathbf{X}(\cdot, i), \mathbf{X}(\cdot, j)) = \frac{\sum_{t=1}^{T} (\mathbf{X}(t, i) - \bar{\mathbf{X}}(\cdot, i))(\mathbf{X}(t, j) - \bar{\mathbf{X}}(\cdot, j))}{\sqrt{\sum_{t=1}^{T} \sigma^2_{\mathbf{X}(\cdot, i)} \sigma^2_{\mathbf{X}(\cdot, j)}}}, \text{ where} \quad (1)$$

$$\bar{\mathbf{X}}(\cdot, k) = \frac{1}{T} \sum_{t=1}^{T} \mathbf{X}(t, k), \text{ and} \quad (2)$$

$$\sigma^2_{\mathbf{X}(\cdot, k)} = \frac{1}{T} \sum_{t=1}^{T} (\mathbf{X}(t, k) - \bar{\mathbf{X}}(\cdot, k))^2 \quad (3)$$

We can generalize this formula to compute time-varying correlations by incorporating a kernel function that takes a time $t$ as input, and returns how much the observed data at each timepoint $\tau \in [-\infty, \infty]$ contributes to the estimated instantaneous correlation[54] at time $t$ (Fig. 9).

Given a kernel function $\kappa_t(\cdot)$ for timepoint $t$, evaluated at timepoints $\tau \in [1, ..., T]$, we can update the static correlation formula in Eq. (1) to estimate the instantaneous correlation at timepoint $t$:

$$\mathrm{timecorr}_{\kappa_t}(\mathbf{X}(\cdot, i), \mathbf{X}(\cdot, j)) = \frac{\sum_{\tau=1}^{T} (\mathbf{X}(\tau, i) - \widetilde{\mathbf{X}}_{\kappa_t}(\cdot, i))(\mathbf{X}(\tau, j) - \widetilde{\mathbf{X}}_{\kappa_t}(\cdot, j))}{\sqrt{\sum_{\tau=1}^{T} \widetilde{\sigma}^2_{\kappa_t}(\mathbf{X}(\cdot, i))\widetilde{\sigma}^2_{\kappa_t}(\mathbf{X}(\cdot, j))}}, \text{ where}$$
$$(4)$$

$$\widetilde{\mathbf{X}}_{\kappa_t}(\cdot, k) = \sum_{\tau=1}^{T} \kappa_t(\tau)\mathbf{X}(\tau, k), \quad (5)$$

$$\widetilde{\sigma}^2_{\kappa_t}(\mathbf{X}(\cdot, k)) = \sum_{\tau=1}^{T} (\mathbf{X}(\tau, k) - \widetilde{\mathbf{X}}_{\kappa_t}(\cdot, k))^2. \quad (6)$$

Here $\mathrm{timecorr}_{\kappa_t}(\mathbf{X}(\cdot, i), \mathbf{X}(\cdot, j))$ reflects the correlation at time $t$ between columns $i$ and $j$ of $\mathbf{X}$, estimated using the kernel $\kappa_t$. We evaluate Eq. (4) in turn for each pair

of columns in $\mathbf{X}$ and for kernels centered on each timepoint in the timeseries, respectively, to obtain a $T$ by $K$ by $K$ timeseries of dynamic correlations, $\mathbf{Y}$. For convenience, we then reshape the upper triangles and diagonals of each timepoint's symmetric correlation matrix into a row vector to obtain an equivalent $T$ by $\left(\frac{K^2 - K}{2} + K\right)$ matrix.

*Dynamic inter-subject functional connectivity.* Equation (4) provides a means of taking a single observation matrix, $\mathbf{X}_n$ and estimating the dynamic correlations from moment to moment, $\mathbf{Y}_{n+1}$. Suppose that one has access to a set of multiple observation matrices that reflect the same phenomenon. For example, one might collect neuroimaging data from several experimental participants, as each participant performs the same task (or sequence of tasks). Let $\mathbf{X}_n^1, \mathbf{X}_n^2, ..., \mathbf{X}_n^P$ reflect the $T$ by $K$ observation matrices ($n = 0$) or reduced correlation matrices ($n > 0$) for each of $P$ participants in an experiment. We can use inter-subject functional connectivity[36,55] (ISFC) to compute the stimulus-driven correlations reflected in the multi-participant dataset at a given timepoint $t$ using:

$$\bar{\mathbf{C}}(t) = M\left(R\left(\frac{1}{2P} \sum_{p=1}^{P} Z(\mathbf{Y}_{n+1}^p(t))^\top + Z(\mathbf{Y}_{n+1}^p(t))\right)\right), \quad (7)$$

where $M$ extracts and vectorizes the upper triangle and diagonal of a symmetric matrix, $Z$ is the Fisher $z$-transformation[56]:

$$Z(r) = \frac{\log(1 + r) - \log(1 - r)}{2}, \quad (8)$$

$R$ is the inverse of $Z$:

$$R(z) = \frac{\exp(2z - 1)}{\exp(2z + 1)}, \quad (9)$$

and $\mathbf{Y}_{n+1}^p(t)$ denotes the correlation matrix at timepoint $t$ (Eqn. (4)) between each column of $\mathbf{X}_n^p$ and each column of the average $\mathbf{X}_n$ from all other participants, $\bar{\mathbf{X}}_n^{\backslash p}$:

$$\bar{\mathbf{X}}_n^{\backslash p} = \frac{1}{P - 1} \sum_{q \in \backslash p} \mathbf{X}_n^q, \quad (10)$$

where $\backslash p$ denotes the set of all participants other than participant $p$. In this way, the $T$ by $\left(\frac{K^2 - K}{2} + K\right)$ DISFC matrix $\bar{\mathbf{C}}$ provides a time-varying extension of the ISFC approach developed by[36].

**Low-dimensional representations of dynamic correlations.** Given a $T$ by $\left(\frac{K^2 - K}{2} + K\right)$ matrix of $n$th-order dynamic correlations, $\mathbf{Y}_n$, we propose two general approaches to computing a $T$ by $K$ low-dimensional representation of those correlations, $\mathbf{X}_n$. The first approach uses dimensionality reduction algorithms to project $\mathbf{Y}_n$ onto a $K$-dimensional space. The second approach uses graph measures to characterize the relative positions of each feature ($k \in [1, ..., K]$) in the network defined by the correlation matrix at each timepoint.

*Dimensionality reduction-based approaches to computing $\mathbf{X}_n$.* The modern toolkit of dimensionality reduction algorithms include Principal Components Analysis[29] (PCA), Probabilistic PCA[57] (PPCA), Exploratory Factor Analysis[58] (EFA), Independent Components Analysis[59,60] (ICA), t-Stochastic Neighbor Embedding[61] (t-SNE), Uniform Manifold Approximation and Projection[62] (UMAP), non-negative matrix factorization[63] (NMF), Topographic Factor Analysis[64] (TFA), Hierarchical Topographic Factor analysis[11] (HTFA), Topographic Latent Source Analysis[65] (TLSA), dictionary learning[66,67], and deep auto-encoders[68], among others. While complete characterizations of each of these algorithms is beyond the scope of the present manuscript, the general intuition driving these approaches is to compute the $T$ by $K$ matrix, $\mathbf{X}$, that is closest to the original $T$ by $J$ matrix, $\mathbf{Y}$, where (typically) $K \ll J$. The different approaches place different constraints on what

properties $\mathbf{X}$ must satisfy and which aspects of the data are compared (and how) in order to optimize how well $\mathbf{X}$ approximates $\mathbf{Y}$.

Applying dimensionality reduction algorithms to $\mathbf{Y}$ yields an $\mathbf{X}$ whose columns reflect weighted combinations (or nonlinear transformations) of the original columns of $\mathbf{Y}$. This has two main consequences. First, with each repeated dimensionality reduction, the resulting $\mathbf{X}_n$ has lower and lower fidelity (with respect to what the "true" $\mathbf{Y}_n$ might have looked like without using dimensionality reduction to maintain tractability). In other words, computing $\mathbf{X}_n$ is a lossy operation. Second, whereas each column of $\mathbf{Y}_n$ may be mapped directly onto specific pairs of columns of $\mathbf{X}_{n-1}$, the columns of $\mathbf{X}_n$ reflect weighted combinations and/or nonlinear transformations of the columns of $\mathbf{Y}_n$. Many dimensionality reduction algorithms are invertible (or approximately invertible). However, attempting to map a given $\mathbf{X}_n$ back onto the original feature space of $\mathbf{X}_0$ will usually require $\mathcal{O}(TK^{2^n})$ space and therefore becomes intractable as $n$ or $K$ grow large.

*Graph measure approaches to computing $\mathbf{X}_n$.* The above dimensionality reduction approaches to approximating a given $\mathbf{Y}_n$ with a lower-dimensional $\mathbf{X}_n$ preserve a (potentially recombined and transformed) mapping back to the original data in $\mathbf{X}_0$. We also explore graph measures that instead characterize each feature's relative position in the broader network of interactions and connections. To illustrate the distinction between the two general approaches we explore, suppose a network comprises nodes $A$ and $B$, along with several other nodes. If $A$ and $B$ exhibit uncorrelated activity patterns, then by definition the functional connection (correlation) between them will be close to 0. However, if $A$ and $B$ each interact with other nodes in similar ways, we might attempt to capture those similarities between $A$'s and $B$'s interactions with those other members of the network.

In general, graph measures take as input a matrix of interactions (e.g., using the above notation, a $K$ by $K$ correlation matrix or binarized correlation matrix reconstituted from a single timepoint's row of $\mathbf{Y}$), and return as output a set of $K$ measures describing how each node (feature) sits within that correlation matrix with respect to the rest of the population. Widely used measures include betweenness centrality (the proportion of shortest paths between each pair of nodes in the population that involves the given node in question[69–73]); diversity and dissimilarity (characterizations of how differently connected a given node is from others in the population[74–76]); eigenvector centrality and pagerank centrality (measures of how influential a given node is within the broader network[77–80]); transfer entropy and flow coefficients (a measure of how much information is flowing from a given node to other nodes in the network[81,82]); $k$-coreness centrality (a measure of the connectivity of a node within its local subgraph[83,84]); within-module degree (a measure of how many connections a node has to its close neighbors in the network[85]); participation coefficient (a measure of the diversity of a node's connections to different subgraphs in the network[85]); and subgraph centrality (a measure of a node's participation in all of the network's subgraphs[86]); among others.

For a given graph measure, $\eta : \mathbb{R}^{K \times K} \rightarrow \mathbb{R}^K$, we can use $\eta$ to transform each row of $\mathbf{Y}_n$ in a way that characterizes the corresponding graph properties of each column. This results in a new $T$ by $K$ matrix, $\mathbf{X}_n$, that reflects how the features reflected in the columns of $\mathbf{X}_{n-1}$ participate in the network during each timepoint (row).

**Dynamic higher-order correlations.** Because $\mathbf{X}_n$ has the same shape as the original data $\mathbf{X}_0$, approximating $\mathbf{Y}_n$ with a lower-dimensional $\mathbf{X}_n$ enables us to estimate high-order dynamic correlations in a scalable way. Given a $T$ by $K$ input matrix, the output of Eq. (4) requires $\mathcal{O}(TK^2)$ space to store. Repeated applications of Eq. (4) (i.e., computing dynamic correlations between the columns of the outputted dynamic correlation matrix) each require exponentially more space; in general the $n$th-order dynamic correlations of a $T$ by $K$ timeseries occupies $\mathcal{O}(TK^{2^n})$ space. However, when we approximate or summarize the output of Eq. (4) with a $T$ by $K$ matrix (as described above), it becomes feasible to compute even very high-order correlations in high-dimensional data. Specifically, approximating the $n$th-order dynamic correlations of a $T$ by $K$ timeseries requires only $\mathcal{O}(TK^2)$ additional space– the same as would be required to compute first-order dynamic correlations. In other words, the space required to store $n + 1$ multivariate timeseries reflecting up to $n$th order correlations in the original data scales linearly with $n$ using our approach (Fig. 8).

**Data.** We examined two types of data: synthetic data and human functional neuroimaging data. We constructed and leveraged the synthetic data to evaluate our general approach[87]. Specifically, we tested how well Eq. (4) could be used to recover known dynamic correlations using different choices of kernel ($\kappa$; Fig. 9), for each of several synthetic datasets that exhibited different temporal properties. We also simulated higher-order correlations and tested how well Eq. (4) could recover these correlations using the best kernel from the previous synthetic data analyses. We then applied our approach to a functional neuroimaging dataset to test the hypothesis that ongoing cognitive processing is reflected in high-order dynamic correlations. We used an across-participant classification test to estimate whether dynamic correlations of different orders contain information about which timepoint in a story participants were listening to.

*Synthetic data: simulating dynamic first-order correlations.* We constructed a total of 400 different multivariate timeseries, collectively reflecting a total of four qualitatively different patterns of dynamic first-order correlations (i.e., 100 datasets reflecting each type of dynamic pattern). Each timeseries comprised 50 features (dimensions) that varied over 300 timepoints. The observations at each timepoint were drawn from a zero-mean multivariate Gaussian distribution with a covariance matrix defined for each timepoint as described below. We drew the observations at each timepoint independently from the draws at all other timepoints; in other words, for each observation $s_t \sim \mathcal{N}(\mathbf{0}, \mathbf{\Sigma}_t)$ at timepoint $t$, $p(s_t) = p(s_t | s_{\backslash t})$.

*Constant*: we generated data with stable underlying correlations to evaluate how Eq. (4) characterized correlation "dynamics" when the ground truth correlations were static. We constructed 100 multivariate timeseries whose observations were each drawn from a single (stable) Gaussian distribution. For each dataset (indexed by $m$), we constructed a random covariance matrix, $\mathbf{\Sigma}_m$:

$$\mathbf{\Sigma}_m = \mathbf{C}\mathbf{C}^\top, \text{ where} \tag{11}$$

$$\mathbf{C}(i, j) \sim \mathcal{N}(0, 1), \text{ and where} \tag{12}$$

$i, j \in [1, 2, ..., 50]$. In other words, all of the observations (for each of the 300 timepoints) within each dataset were drawn from a multivariate Gaussian distribution with the same covariance matrix, and the 100 datasets each used a different covariance matrix.

*Random*: we generated a second set of 100 synthetic datasets whose observations at each timepoint were drawn from a Gaussian distribution with a new randomly constructed (using Eq. (11)) covariance matrix. Because each timepoint's covariance matrix was drawn independently from the covariance matrices for all other timepoints, these datasets provided a test of reconstruction accuracy in the absence of any meaningful underlying temporal structure in the dynamic correlations underlying the data.

*Ramping*: we generated a third set of 100 synthetic datasets whose underlying correlations changed gradually over time. For each dataset, we constructed two "anchor" covariance matrices using Eq. (11), $\mathbf{\Sigma}_{\text{start}}$ and $\mathbf{\Sigma}_{\text{end}}$. For each of the 300 timepoints in each dataset, we drew the observations from a multivariate Gaussian distribution whose covariance matrix at each timepoint $t \in [0, ..., 299]$ was given by

$$\mathbf{\Sigma}_t = \left(1 - \frac{t}{299}\right)\mathbf{\Sigma}_{\text{start}} + \frac{t}{299}\mathbf{\Sigma}_{\text{end}}. \tag{13}$$

The gradually changing correlations underlying these datasets allow us to evaluate the recovery of dynamic correlations when each timepoint's correlation matrix is unique (as in the random datasets), but where the correlation dynamics are structured and exhibit first-order autocorrelations (as in the constant datasets).

*Event*: we generated a fourth set of 100 synthetic datasets whose underlying correlation matrices exhibited prolonged intervals of stability, interspersed with abrupt changes. For each dataset, we used Eq. (11) to generate five random covariance matrices. We constructed a timeseries where each set of 60 consecutive samples was drawn from a Gaussian with the same covariance matrix. These datasets were intended to simulate a system that exhibits periods of stability punctuated by occasional abrupt state changes.

*Synthetic data: simulating dynamic high-order correlations.* We developed an iterative procedure for constructing timeseries data that exhibits known dynamic high-order correlations. The procedure builds on our approach to generating dynamic first-order correlations. Essentially, once we generate a timeseries with known first-order correlations, we can use the known first-order correlations as a template to generate a new timeseries of second-order correlations. In turn, we can generate a timeseries of third-order correlations from the second-order correlations, and so on. In general, we can generate order $n$ correlations from a timeseries of order $n - 1$ correlations, for any $n > 1$. Finally, given the order $n$ timeseries, we can reverse the preceding process to generate an order $n - 1$ timeseries, an order $n - 2$ order timeseries, and so on, until we obtain an order 0 timeseries of simulated data that reflects the chosen high-order dynamics.

The central mathematical operation in our procedure is the Kronecker product ($\otimes$). The Kronecker product of a $K \times K$ matrix, $m_1$, with itself (i.e., $m_1 \otimes m_1$) produces a new $K^2 \times K^2$ matrix, $m_2$ whose entries reflect a scaled tiling of the entries in $m_1$. If these tilings (scaled copies of $m_1$) are indexed by row and column, then the tile in the $i$th row and $j$th column contains the entries of $m_1$, multiplied by $m_1(i, j)$. Following this pattern, the Kronecker product $m_2 \otimes m_2$ yields the $K^4 \times K^4$ matrix $m_3$ whose tiles are scaled copies of $m_2$. In general, repeated applications of the Kronecker self-product may be used to generate $m_{n+1} = m_n \otimes m_n$ for $n > 1$, where $m_{n+1}$ is a $K^{2^n} \times K^{2^n}$ matrix. After generating a first-order timeseries of dynamic correlations (see Synthetic data: simulating dynamic first-order correlations), we use this procedure (applied independently at each timepoint) to transform it into a timeseries of $n$th-order correlations. When $m_{n+1}$ is generated in this way, the temporal structure of the full timeseries (i.e., constant, random, ramping, event) is preserved, since changes in the original first-order timeseries are also reflected in the scaled tilings of itself that comprise the higher-order matrices.

Given a timeseries of $n$th-order correlations, we then need to work "backwards" in order to generate the order-zero timeseries. If the $n$th-order correlation matrix at a given timepoint is $m_n$, then we can generate an order $n - 1$ correlation matrix (for $n > 1$) by taking a draw from $\mathcal{N}(0, m_n)$ and reshaping the resulting vector to

have square dimensions. To force the resulting matrix to be symmetric, we remove its lower triangle, and replace the lower triangle with (a reflected version of) its upper triangle. Intuitively, the reshaped matrix will look like a noisy (but symmetric) version of the template matrix, $m_{n-1}$. (When $n = 1$, no reshaping is needed; the resulting $K$-dimensional vector may be used as the observation at the given timepoint.) After independently drawing each timepoint's order $n - 1$ correlation matrix from that timepoint's order $n$ correlation matrix, this process can be applied repeatedly until $n = 0$. This results in a $K$-dimensional timeseries of $T$ observations containing the specified high-order correlations at orders 1 through $n$. Following our approach to generating synthetic data exhibiting known first-order correlations, we constructed a total of 400 additional multivariate timeseries, collectively reflecting a total of four qualitatively different patterns of dynamic correlations (i.e., 100 datasets reflecting each type of dynamic pattern: constant, random, ramping, and event). Each timeseries comprised 10 zero-order features (dimensions) that varied over 300 timepoints. After applying our dynamic correlation estimation procedure, this yielded a 100-dimensional timeseries of first-order features that could then be used to estimate dynamic second-order correlations. (We chose to use $K = 10$ zero-order features for our higher-order simulations in order to put the accuracy computations displayed in Figs. 2 and 3 on a roughly even footing.)

*Functional neuroimaging data collected during story listening.* We examined an fMRI dataset collected by[36] that the authors have made publicly available at arks.princeton.edu/ark:/88435/dsp015d86p269k. The dataset comprises neuroimaging data collected as participants listened to an audio recording of a story (intact condition; 36 participants), listened to temporally scrambled recordings of the same story (17 participants in the paragraph-scrambled condition listened to the paragraphs in a randomized order and 36 in the word-scrambled condition listened to the words in a randomized order), or lay resting with their eyes open in the scanner (rest condition; 36 participants). Full neuroimaging details may be found in the original paper for which the data were collected[36]. Procedures were approved by the Princeton University Committee on Activities Involving Human Subjects, and by the Western Institutional Review Board (Puyallup, WA). All subjects were native English speakers with normal hearing and provided written informed consent.

*Hierarchical topographic factor analysis (HTFA):* following our prior related work, we used HTFA[11] to derive a compact representation of the neuroimaging data. In brief, this approach approximates the timeseries of voxel activations (44,415 voxels) using a much smaller number of radial basis function (RBF) nodes (in this case, 700 nodes, as determined by an optimization procedure[11]). This provides a convenient representation for examining full-brain network dynamics. All of the analyses we carried out on the neuroimaging dataset were performed in this lower-dimensional space. In other words, each participant's data matrix, $\mathbf{X}_0$, was a number-of-timepoints by 700 matrix of HTFA-derived factor weights (where the row and column labels were matched across participants). Code for carrying out HTFA on fMRI data may be found as part of the BrainIAK toolbox[88], which may be downloaded at brainiak.org.

**Temporal decoding.** We sought to identify neural patterns that reflected participants' ongoing cognitive processing of incoming stimulus information. As reviewed by Simony et al.[36], one way of homing in on these stimulus-driven neural patterns is to compare activity patterns across individuals (e.g., using ISFC analyses). In particular, neural patterns will be similar across individuals to the extent that the neural patterns under consideration are stimulus-driven, and to the extent that the corresponding cognitive representations are reflected in similar spatial patterns across people[55]. Following this logic, we used an across-participant temporal decoding test developed by[11] to assess the degree to which different neural patterns reflected ongoing stimulus-driven cognitive processing across people (Fig. 10). The approach entails using a subset of the data to train a classifier to decode stimulus timepoints (i.e., moments in the story participants listened to) from neural patterns. We use decoding (forward inference) accuracy on held-out data, from held-out participants, as a proxy for the extent to which the inputted neural patterns reflected stimulus-driven cognitive processing in a similar way across individuals.

*Forward inference and decoding accuracy.* We used an across-participant correlation-based classifier to decode which stimulus timepoint matched each timepoint's neural pattern (Fig. 10). We first divided the participants into two groups: a template group, $\mathcal{G}_{template}$ (i.e., training data), and a to-be-decoded group, $\mathcal{G}_{decode}$ (i.e., test data). We used Eq. (7) to compute a DISFC matrix for each group ($\bar{\mathbf{C}}_{template}$ and $\bar{\mathbf{C}}_{decode}$, respectively). We then correlated the rows of $\bar{\mathbf{C}}_{template}$ and $\bar{\mathbf{C}}_{decode}$ to form a number-of-timepoints by number-of-timepoints decoding matrix, $\mathbf{\Lambda}$. In this way, the rows of $\mathbf{\Lambda}$ reflected timepoints from the template group, while the columns reflected timepoints from the to-be-decoded group. We used $\mathbf{\Lambda}$ to assign temporal labels to each row $\bar{\mathbf{C}}_{decode}$ using the row of $\bar{\mathbf{C}}_{template}$ with which it was most highly correlated. We then repeated this decoding procedure, but using $\mathcal{G}_{decode}$ as the template group and $\mathcal{G}_{template}$ as the to-be-decoded group. Given the true timepoint labels (for each group), we defined the decoding accuracy as the average proportion of correctly decoded timepoints, across both groups. We

defined the relative decoding accuracy as the difference between the decoding accuracy and chance accuracy (i.e., $\frac{1}{T}$).

*Feature weighting and testing.* We sought to examine which types of neural features (i.e., activations, first-order dynamic correlations, and higher-order dynamic correlations) were informative to the temporal decoders. Using the notation above, these correspond to $\mathbf{X}_0, \mathbf{X}_1, \mathbf{X}_2, \mathbf{X}_3$, and so on.

One challenge to fairly evaluating high-order correlations is that if the kernel used in Eq. (4) is wider than a single timepoint, each repeated application of the equation will result in further temporal blur. Because our primary assessment metric is temporal decoding accuracy, this unfairly biases against detecting meaningful signal in higher-order correlations (relative to lower-order correlations). We attempted to mitigate temporal blur in estimating each $\mathbf{X}_n$ by using a Dirac $\delta$ function kernel (which places all of its mass over a single timepoint; Figs. 9b and 10a) to compute each lower-order correlation ($\mathbf{X}_1, \mathbf{X}_2, ..., \mathbf{X}_{n-1}$). We then used a new (potentially wider, as described below) kernel to compute $\mathbf{X}_n$ from $\mathbf{X}_{n-1}$. In this way, temporal blurring was applied only in the last step of computing $\mathbf{X}_n$. We note that, because each $\mathbf{X}_n$ is a low-dimensional representation of the corresponding $\mathbf{Y}_n$, the higher-order correlations we estimated reflect true correlations in the data with lower fidelity than estimates of lower-order correlations. Therefore, even after correcting for temporal blurring, our approach is still biased against finding meaningful signal in higher-order correlations.

After computing each $\mathbf{X}_1, \mathbf{X}_2, ..., \mathbf{X}_{n-1}$ for each participant, we divided participants into two equally sized groups ($\pm 1$ for odd numbers of participants): $\mathcal{G}_{train}$ and $\mathcal{G}_{test}$. We then further subdivided $\mathcal{G}_{train}$ into $\mathcal{G}_{train_1}$ and $\mathcal{G}_{train_2}$. We then computed $\mathbf{\Lambda}$ (temporal correlation) matrices for each type of neural feature, using $\mathcal{G}_{train_1}$ and $\mathcal{G}_{train_2}$. This resulted in $n + 1 \mathbf{\Lambda}$ matrices (one for the original timeseries of neural activations, and one for each of $n$ orders of dynamic correlations). Our objective was to find a set of weights for each of these $\mathbf{\Lambda}$ matrices such that the weighted average of the $n + 1$ matrices yielded the highest decoding accuracy. We used quasi-Newton gradient ascent[89], using decoding accuracy (for $\mathcal{G}_{train_1}$ and $\mathcal{G}_{train_2}$) as the objective function to be maximized, to find an optimal set of training data-derived weights, $\phi_{0,1,...,n}$, where $\sum_{i=0}^{n} \phi_i = 1$ and where $\phi_i \geq 0 \forall i \in [0, 1, ..., n]$.

After estimating an optimal set of weights, we computed a new set of $n + 1 \mathbf{\Lambda}$ matrices correlating the DISFC patterns from $\mathcal{G}_{train}$ and $\mathcal{G}_{test}$ at each timepoint. We use the resulting decoding accuracy of $\mathcal{G}_{test}$ timepoints (using the weights in $\phi_{0,1,...,n}$ to average the $\mathbf{\Lambda}$ matrices) to estimate how informative the set of neural features containing up to $n$th order correlations were.

We used a permutation-based procedure to form stable estimates of decoding accuracy for each set of neural features. In particular, we computed the decoding accuracy for each of 10 random group assignments of $\mathcal{G}_{train}$ and $\mathcal{G}_{test}$. We report the mean accuracy (along with 95% confidence intervals) for each set of neural features.

*Identifying robust decoding results.* The temporal decoding procedure we use to estimate which neural features support ongoing cognitive processing is governed by several parameters. In particular, Eq. (4) requires defining a kernel function, which can take on different shapes and widths. For a fixed set of neural features, each of these parameters can yield different decoding accuracies. Further, the best decoding accuracy for a given timepoint may be reliably achieved by one set of parameters, whereas the best decoding accuracy for another timepoint might be reliably achieved by a different set of parameters, and the best decoding accuracy across all timepoints might be reliably achieved by still another different set of parameters. Rather than attempting to maximize decoding accuracy, we sought to discover the trends in the data that were robust to classifier parameters choices. Specifically, we sought to characterize how decoding accuracy varied (under different experimental conditions) as a function of which neural features were considered.

To identify decoding results that were robust to specific classifier parameter choices, we repeated our decoding analyses after substituting into Eq. (4) each of a variety of kernel shapes and widths. We examined Gaussian (Fig. 9c), Laplace (Fig. 9d), and Mexican Hat (Fig. 9e) kernels, each with widths of 5, 10, 20, and 50 samples. We then report the average decoding accuracies across all of these parameter choices. This enabled us to (partially) factor out performance characteristics that were parameter-dependent, within the set of parameters we examined.

*Reverse inference.* The dynamic patterns we examined comprise high-dimensional correlation patterns at each timepoint. To help interpret the resulting patterns in the context of other studies, we created summary maps by computing the across-timepoint average pairwise correlations at each order of analysis (first order, second order, etc.). We selected the 10 strongest (absolute value) correlations at each order. Each correlation is between the dynamic activity patterns (or patterns of dynamic high-order correlations) measured at two RBF nodes (see Hierarchical Topographic Factor Analysis). Therefore, the 10 strongest correlations involved up to 20 RBF nodes. Each RBF defines a spatial function whose activations range from 0 to 1. We constructed a map of RBF components that denoted the endpoints of the 10 strongest correlations (we set each RBF to have a maximum value of 1). We then carried out a meta analysis using Neurosynth[38] to identify the 10 terms most

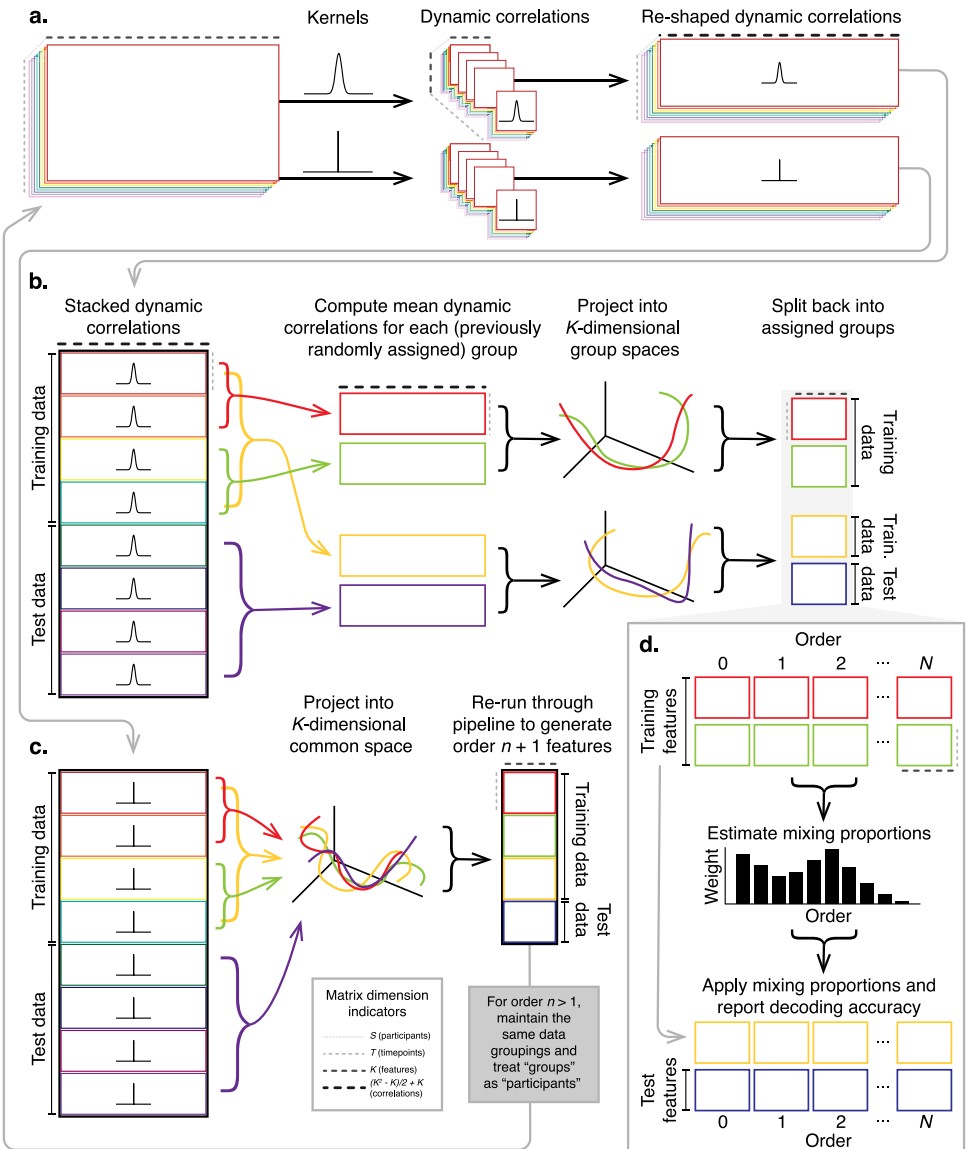

**Fig. 10 Decoding analysis pipeline. a. Computing dynamic correlations from timeseries data.** Given a timeseries of observations as a $T \times K$ matrix (or a set of $S$ such matrices), we use Equation (4) to compute each participant's DISFC (relative to other participants in the training or test sub-group, as appropriate). We repeat this process twice-- once using the analysis kernel (shown here as a Gaussian in the upper row of the panel), and once using a $\delta$ function kernel (lower row of the panel). **b. Projecting dynamic correlations into a lower-dimensional space**. We project the training and test data into $K$-dimensional spaces to create compact representations of dynamic correlations at the given order (estimated using the analysis kernel). **c. Kernel trick**. We project the dynamic correlations computed using a $\delta$ function kernel into a common $K$-dimensional space. These low-dimensional embeddings are fed back through the analysis pipeline in order to compute features at the next-highest order. **d. Decoding analysis**. We split the training data into two equal groups, and optimize the feature weights (i.e., dynamic correlations at each order) to maximize decoding accuracy. We then apply the trained classifier to the (held-out) test data.

commonly associated with the given map. This resulted in a set of 10 terms associated with the average dynamic correlation patterns at each order.

**Reporting summary**. Further information on research design is available in the Nature Research Reporting Summary linked to this article.

## Data availability
The authors declare that the data supporting the findings of this study as well as the source data for this paper are available at github.com/ContextLab/timecorr-paper/releases/tag/v0.4 and has been deposited in the Zenodo database under accession code https://doi.org/10.5281/zenodo.5165253. The source data underlying Figs. 2–6 and Supplementary Figs. S1–S9 are provided as Source Data files. Source Data are provided with the manuscript. The raw fMRI data are protected and are not available due to data privacy laws. The processed fMRI dataset collected by[36] has been made publicly available[90] at arks.princeton.edu/ark:/88435/dsp015d86p269k. Source data are provided with this paper.

## Code availability
All of our analysis code may be downloaded from github.com/ContextLab/timecorr-paper/releases/tag/v0.4. We have also published a companion Python toolbox that may be downloaded from timecorr.readthedocs.io.

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

## Acknowledgements

We acknowledge discussions with Luke Chang, Vassiki Chauhan, Hany Farid, Paxton Fitzpatrick, Andrew Heusser, Eshin Jolly, Aaron Lee, Qiang Liu, Matthijs van der Meer, Judith Mildner, Gina Notaro, Stephen Satterthwaite, Emily Whitaker, Weizhen Xie, and Kirsten Ziman. Our work was supported in part by NSF EPSCoR Award Number 1632738 to J.R.M. and by a sub-award of DARPA RAM Cooperative Agreement N66001-14-2-4-032 to J.R.M. The content is solely the responsibility of the authors and does not necessarily represent the official views of our supporting organizations.

## Author contributions

Concept: J.R.M. Implementation: T.H.C., L.L.W.O., and J.R.M. Analyses: L.L.W.O. and J.R.M. Writing: L.L.W.O. and J.R.M.

## Competing interests

The authors declare no competing interests.
