## [Peer Review File · Nature Communications]

High-level cognition during story listening is reflected in high-order dynamic correlations in neural activity patternsReviewers' comments:

Reviewer #1 (Remarks to the Author):

This paper proposes a novel approach to quantifying "deep structure" within fMRI data that increases in complexity with complex perceptual inputs, and maps meaningfully onto known cortical hierarchies.

The paper is well written and explores an intriguing line of thread. Some readers might struggle a little with the meaning of "higher order time-dependent correlations between subjects" but I think the authors do an excellent job in unpicking the semantics of these concepts.

As with all new measures, there are a few issues with the validation of its sensitivity, specificity which I explain below.

Major

1. The validation analyses (figure 4) are reassuring for the detection of non-stationarities in the cross correlation structure of synthetic time series, but I think they lack in three respects:
(1) Empirical BOLD time series (such as those analysed in the empirical section) have strong auto-correlations; therefore transient changes in the cross correlations are not isolated to the time of occurrence but are forward propagated in time due to auto-correlations – this could/should be addressed with at least one exemplar analysis.
(2) Much of the paper is devoted to higher order correlations, not time dependent correlations: these examples only insert changes in first order correlations. What are the origins of putative second and third order correlations that form the conceptual basis of the paper? In dynamic systems theory, these typically arise due to the presence of nonlinearities (such as chaos) and non stationarities due to more complex nonlinear dynamics such as multi-stability (see [1] for examples and code for simulation) – Ideally these would also be validated through the use of simulated nonlinear dynamics.
(3) On the same point, since these synthetic data contain only dynamic first order correlations, the behaviour of the higher order vectors X should be near null and this should be explored.

In addition, testing the algorithm on a block-design task fMRI data set could be informative in recovering known discrete change points.

2. It is not clear what the analyses with the "normalized accuracy" (Fig 5 RHS) show – it seems here that the resting state analyses can be decoded quite well with low order correlations – but why normalize to have unit maximum? Doesn't this artificially inflate those results? Why not use a statistical approach such as permutation or bagging?

3. Related to these preceding concerns, the presence of non-stationarities in low order correlations will likely propagate to apparent (spurious) changes in higher order correlations. Therefore, I suggest the authors use a resampling scheme in the Fourier [2] or wavelet [3] domain and benchmark their empirical findings against these null (which preserve all trivial auto-and cross-correlations but destroy higher order ones). This may be particularly helpful in explaining the presence of low order inter-subject correlations in the resting state data (Fig. 6, right hand column).

Minor;

Abstract:

1. Isn't the first sentence of the abstract a bit tautological? Our thoughts arise from neural things that change with our thoughts ...??
2. The "brain's connectome" refers to the structural connections of the brain [6] – not the changes in functional connectivity referred to here

Introduction:

1. The notion that cognition is mediated by dynamic interactions between brain structures is much older than these cited references suggest – e.g. [4,5] but many others].

Methods:

1. Line 91: Don't you mean "(T by $((K^2-K)/2+K)$ -dimensional)"?
2. The discussion of dimensionality reduction on lines 127-148 is interesting but it's not strictly required for the paper and could be deleted. Likewise, for the discussion of graph metrics, lines 158-175.

Discussion:

1. Is figure 8 schematic only? That is not evident from the text or caption.

References:

- 1: Heitmann, S., & Breakspear, M. (2018). Putting the "dynamic" back into dynamic functional connectivity. *Network Neuroscience*, 2(02), 150-174.
- 2: Prichard D, Theiler J (1994) Generating surrogate data for time series with several simultaneously measured variables. *Physical Review Letters*, 73:951-954.
- 3: Breakspear, M., Brammer, M. J., Bullmore, E. T., Das, P., & Williams, L. M. (2004). Spatiotemporal wavelet resampling for functional neuroimaging data. *Human brain mapping*, 23(1), 1-25.
- 4: Friston, K. J. (2000a) The labile brain. I. Neuronal transients and nonlinear coupling. *Phil. Trans. Roy. Soc. Lon.*, 355B, 215–236.
- 5: Grossberg, S. (1988). Nonlinear neural networks: Principles, mechanisms, and architectures. *Neural networks*, 1(1), 17-61.
- 6: Sporns1: Sporns, O., Tononi, G., & Kötter, R. (2005). The human connectome: a structural description of the human brain. *PLoS computational biology*, 1(4), e42.

Signed, Michael Breakspear

Reviewer #2 (Remarks to the Author):

In this manuscript, Owen et al. introduce and test a set of methods for analyzing neural dynamics across multiple "orders" of co-variation within and across brains. In particular, they examine "high order dynamic correlations", which are brain subnetworks which exhibit correlated changes across individuals. They apply their new method to brain dynamics from resting conditions, and also from task data in which participants listened to a narrative stimulus scrambled at the scales of words, sentences and paragraphs. Overall, they conclude that the higher-order correlation patterns contain more useful information for decoding the timepoints of the more "cognitive rich" stimulus states, whereas the lower-order correlation patterns are ideal for decoding brain states associated with more basic, stimulus driven representations.

Overall, the concept is extremely intriguing and the paper is fairly well-written. At the same time, more work will be needed, both in simulation and also in the empirical analyses, in order to substantiate the central claims, and clarify the logic of some of the analyses.

MAIN POINTS

[1] The simulations used to validate the methodology are only used to test different kernel widths; however, the most innovative aspect of the paper is the recursive "higher order" correlation method, illustrated in Figure 1. The authors should conduct simulations to test (i) whether the higher-order correlation methods are indeed picking up on the structures (e.g. graph-state covariations) that are described visually in Figure 1, and (ii) that this translates into "optimal" temporal decoding of higher-order covariation structure information from higher order correlations. Such simulations could provide a basic validity support for the key logic that is applied to the empirical data in Figures 4-6.

[2] In the original empirical analysis by Simony et al, the number of "reliably responding" voxels and "reliably inter-subject correlated" voxels for these data was shown to increase across conditions, with "word scramble" producing the smallest number of reliable voxels, and "intact" providing the largest. Could it be the case that the "higher order correlation" method does better when there is a larger number of inter-subject reliable voxels? Could the total number of reliable voxels per condition be confounded with the "optimal order" for decoding at that level? It seems that this issue could be directly resolved using simulation.

[3] In Figure 5, it appears that "Rest" has a "normalized accuracy" score greater than 80%, and this is a little jarring. The text does not specify how the accuracy normalization is performed. Is it divisive? Please specify precisely. It is difficult to understand what one is supposed to take away from Figures 5B and 5D. The error ribbons appear to overlap for all curves except "Intact". In fact, the "rest" decoding appears to numerically exceed most conditions. Related to this, the text

[4] The primary results are generated by averaging across kernel widths. I understand that the authors do not wish to share all of the complexity that comes with high-dimensional analyses..... but at the same time, the averaging across kernel widths leaves the reader with the feeling that the main results vary substantially across kernel widths. Is it possible to report for which specific kernel widths the primary results (intact condition better decoded by higher order correlations) will hold?

[5] Suppose that the decoding target was relaxed in time so that, instead of decoding a specific TR, the task was instead to decode (say) a specific 10-15 second window. Might this resolve some of the variability in the results related to different kernels? In particular, it may be more difficult for very temporally coarse kernels to succeed in decoding temporally precise targets? But if we make the target of decoding less temporally specific, then perhaps a wider range of kernels could produce results that are most consistent with one another?

[6] line 367: "This dataset provided a convenient means of testing our hypothesis that different levels of cognitive processing and engagement are supported by different orders of brain activity dynamics." — Please reconsider the use of the phrase "supported by" here. Since the data are analyzed using a decoding analysis, it is unclear whether we could infer whether particular brain dynamics are causal in supporting particular cognitive states. If the authors want to make this causal claim, they should buttress the causal claim with argument.

[7] I must confess that I had great difficulty understanding what is shown in Figure 7, to the extent that I cannot really comment on the results at all. The methods state: "We constructed a map of RBF components that denoted the endpoints of the 10 strongest correlations (we set each RBF to have a maximum value of 1)." But I cannot understand what is meant by an "endpoint" for correlations of (say) order 3. Should we expect this endpoint to correspond simply to a single region? Single regions are indeed plotted in the left-hand side of Figure 7. But, I had thought that the higher order correlations no longer involved single regions? Moreover, the methods that: "However, attempting to map a given X_n back onto the original feature space of X_0 will usually require $O(K^2n)$ space and therefore becomes intractable as n or K grow large." If it is not possible to map back from the higher order correlations to the original feature space, then I am a

bit confused as to how the zero-order and higher-order correlation "endpoints" can be plotted on the same brain surface in Figure 7. Can the "endpoint" of a 4th-order correlation still be a single brain region?

SMALL POINTS

— Figure 4 contains "gray" markings in the legend, for narrow kernels, but there does not appear to be any gray coloring at all in the figures; since it only appears to be the Gaussian kernel whose width is varied in the figures (?), why not provide varying saturations of green to indicate changing kernel widths?

— Figure 4 should contain a "ground truth" curve, indicating when exactly the event transitions / ramps /decays occurred in the underlying ground-truth covariance matrices. This will enable a more direct visual comparison between the timing of the simulation and the recovered time-courses from the analysis.

— The colormap for the t-values in Figure 6 is confusing... in particular, small positive t-values are shown in pale blue, while negative t-values are shown in dark blue. Am I missing something? Is the white color supposed to correspond to $t = 0$? Surely it would make sense to use a colormap that is in perceptually symmetric around zero?

— What is "decoded" during the resting scan? Please clarify that, just as with the task-locked scans, the "timepoint" is what is decoded during rest. This would be consistent with the interpretation of decoding performance in terms of time-varying changes in arousal and engagement over the course of the rest period.

— How does the dimensionality of the data input to the PCA vary across "orders" of the analysis? In other words, if one were to plot the % variance explained as a function of number of principal components, how does the shape of this curve vary as one iterates from lower-order to higher-order correlations? Do the dynamics become more "low-dimensional" / compressible at higher orders?

— "which types of neural activity patterns might comprise the neural code" ... this might read better as "which types of neural activity patterns might compose the neural code"
(The code comprises patterns, but the patterns compose the code)

— there are some very long "methods review" paragraphs (e.g. lines 161-174) in which long lists of related methods are cited. Since this is not a review paper, it is not clear that such long lists, with little commentary, belong in the main text of an empirical article. Is it possible to trim these lists to discuss only the most relevant related work?

Reviewer #3 (Remarks to the Author):

Owen et al. develop a novel method for estimating higher-order correlations in fMRI activity, and validate it using simulated data. They then characterize higher-order correlation patterns in fMRI data collected as people listened to intact and scrambled stories and rested, and found that classifiers performed better for predicting the intact story timepoint when they used higher-order correlation features. In contrast, classifiers performed better for predicting time for the scrambled and rest condition when they only included activity or first-order correlation patterns.

Although the paper represents a helpful methodological advance in that it will allow researchers to characterize higher-order correlations quickly and efficiently, the robustness of the classification results and theoretical advance is less clear. My main comments are described in detail below.

Major comments:

1. The primary conclusion of the paper is that "high-level cognition is supported by high-order brain network dynamics." "High-level cognition", however, is never defined, and sometimes the

intact story is said to reflect “cognitive engagement”, again a vague term. Is the claim that higher-order correlations reflect and/or support cognitive processes related to story understanding, like attention, learning, and memory, but not lower-order processes like perception? Clearly defining the process or processes being reflected here would increase the impact of the work.

2. Related to the first point about definitions, I think a longer discussion in the introduction about high-order correlations would be helpful. Is there an intuitive way to understand correlations beyond 2nd order? Is there any theoretical reason to look at correlations between order 0 and 10 but not beyond? Although I recognize the authors’ sophisticated and rigorous methodological work I am still struggling to understand what these high-order correlations represent.

3. The paper’s main conclusion relies on qualitative comparison of decoding accuracy models with higher-order correlation features for the story condition but lower-order correlation features for the rest condition. However, it is hard to interpret how meaningful these differences are, especially since Figure 5 (left panel) appears to show stable accuracy regardless of the feature orders included in the model for all scan conditions. Are differences in model accuracy as a function of correlation orders included statistically significant? Are they consistent across additional splits of the data? (Currently there are only 10, and based on Figure 5 there appears to be high variability across splits.) If not, the broad conclusion that higher-order feature models are better for the story condition but lower-order features are better for scrambled stories and rest may not be justified.

4. The article argues that high-order correlations contribute to predictive power during an intact story condition but not at rest. However, the bottom right matrix in Figure 6 perform less well for classifying timepoint than higher-order correlations (up to order 5). This appears counter to the argument that lower-order features are better for classifying time point during rest. Is this interpretation warranted?

5. Given that head motion can be a significant confound in functional connectivity (and especially dynamic functional connectivity studies), are high-order correlation patterns affected by data quality or quantity (e.g., number of frames available, head motion)? Could these differences explain the differential contribution of different correlation orders to decoding accuracy across scan conditions? For example, could subjects have moved their heads more—or more consistently—during rest than stories, or vice versa? Are estimates of high-order correlations systematically affected by time series length, and did scan lengths differ across conditions?

6. The paper’s methods are complex, and include defining network nodes with HTFA, estimating dynamic functional connectivity with different kernels, projecting these patterns in low-dimensional space using two approaches, repeating this process for up to 10th-order correlations, temporal decoding with the resulting matrices after a parameter search to optimize nth-order matrix weights, averaging model performance across 12 classifiers with different combinations of kernels and kernel widths. Especially in light of this complexity the figures are a good opportunity to walk readers through the analysis pipeline. However, the current figures are a bit hard to follow. For example, Figure 1 does a nice job explaining the difference between fMRI activity vs. functional connectivity vs. network-to-network interactions. However, it is difficult to extract the most relevant information for understanding the rest of the paper from this graph. Is it necessary to distinguish within- from across-brain relationships for the current work? Univariate vs. multivariate relationships? What exactly is meant by a “homology”?

Similarly, the take home messages are not easy to extract from Figures 5 and 6. Figure 5 shows minimal differences between classifiers that include different order features, thus suggesting that decoding is relatively stable within a scan condition regardless of the features used, a point contrary to the main message of the paper.

Likewise, it took me several close reads of the Figure 6 caption to understand the organization and color scale, and I’m still not sure I could replicate how the value in each cell was calculated. Perhaps a schematic would help here? Or a step-by-step explanation? Some summary take-aways could also help too. For example, does the consistent blue pattern in the top left matrix show that lower-order features are always worse than higher-order features, and does the all-red pattern in the second matrix in the bottom row show that lower-order features are always better than

higher-order features? The weights on the diagonal are a bit confusing in light of the manuscript's claims about the value of higher-order correlations for decoding scan time point, too. Are results associated with correlations of orders greater than 4 meaningful or important given that their weights in predictive models are, on average, 0?

Minor comments:

1. Did the hierarchical topographic factor analysis (HTFA) approach used for node definition compromise the independence between training and test data? If all subjects were used to define the regions of interest then the test data would have influenced the features used at training. A completely independent approach would define the nodes in the training set and apply them to the test set, or use a pre-defined functional brain atlas developed in an independent sample.

2. Related to HTFA, Manning et al. (2018) previously shown that dynamic functional connectivity patterns examined with this approach can predict time points of an intact and scrambled story using the same data reported here. It would be helpful to acknowledge this prior work and clarify what the current work adds.

3. It is worth highlighting in the text that, in contrast to the majority of applications of machine learning approaches to fMRI, the decoding results do not provide predictions for individual subjects, but instead predict time points based on group-averaged data.

4. The concluding remarks section is a bit overstated. (It reads: "The complex hierarchy of dynamic interactions that underlie our thoughts is perhaps the greatest mystery in modern science. Methods for characterizing the dynamics of high-order correlations in neural data provide a window into the neural basis of cognition. By showing that high-level cognition is reflected in high-order network dynamics, we have elucidated the next step on the path towards understanding the neural basis of cognition.") Although understanding the neural bases of cognition is certainly an important goal of science, the broad claims here almost undermine the authors' interesting work by "protesting too much".

We appreciate the reviewers' comments, and have included point-by-point responses to each of the three reviewers in the "response to referees" document attached to our resubmission. The reviewers' comments are shown in *italics* and our responses are shown in **bold**.

Reviewer #1 (Remarks to the Author):

This paper proposes a novel approach to quantifying "deep structure" within fMRI data that increases in complexity with complex perceptual inputs, and maps meaningfully onto known cortical hierarchies.

The paper is well written and explores an intriguing line of thread. Some readers might struggle a little with the meaning of "higher order time-dependent correlations between subjects" but I think the authors do an excellent job in unpicking the semantics of these concepts.

As with all new measures, there are a few issues with the validation of its sensitivity, specificity which I explain below.

Major

1. The validation analyses (figure 4) are reassuring for the detection of non-stationarities in the cross correlation structure of synthetic time series, but I think they lack in three respects:

(1) Empirical BOLD time series (such as those analysed in the empirical section) have strong auto-correlations; therefore transient changes in the cross correlations are not isolated to the time of occurrence but are forward propagated in time due to auto-correlations – this could/should be addressed with at least one exemplar analysis.

We agree that real neural data (including BOLD data) have many complexities beyond what we have incorporated into our simulations. Autocorrelations like what the reviewer is referring to are one such property; others include anatomical and physiological constraints, dynamics of brain patterns that support cognition, measurement properties, and so on. Our synthetic datasets featured in Figures 2A, 2C, and 3 (leftmost panel, third panel from the left) exhibit first-order autocorrelations of the sort described by the reviewer, and the synthetic datasets featured in Figures 2D and 3 (rightmost panel) exhibit first-order autocorrelations that are punctuated by event boundaries. We have added a clarifying note to this effect on page 22.

Despite imposing autocorrelations on the synthetic datasets, we feel that even stronger evidence that our approach and key findings are not solely a reflection of autocorrelation come from our analyses of BOLD data (e.g. Figs. 4, 5, and 6 in the revised manuscript). Specifically, the dynamic high-order correlations recovered by our approach should only be able to precisely decode temporal information to the extent that our recovery algorithm is robust to non-cognitively driven autocorrelations and other potential artifacts in the BOLD data.

(2) Much of the paper is devoted to higher order correlations, not time dependent correlations: these examples only insert changes in first order correlations. What are the origins of putative second and third order correlations that form the conceptual basis of the paper? In dynamic systems theory, these typically arise due to the presence of nonlinearities (such as chaos) and non stationarities due to more complex nonlinear dynamics such as multi-stability (see [1] for examples and code for simulation) – Ideally these would also be validated through the use of simulated nonlinear dynamics.

First, we have added a new set of simulations of high-order dynamic correlations (e.g., Fig. 3) that demonstrate that we are able to accurately recover high-order dynamic correlations from synthetic data.

The reviewer's question about the origins of high-order correlations is important, but our analyses do not speak to the physiological basis of these phenomena. Rather, our approach here is to characterize the circumstances under which high-order correlation dynamics reflect processing of richly structured stimuli (e.g. Figs. 4, 5, 6) and which brain regions participate in these dynamics (e.g. Figs. 6, 7, S3, S4, S5, S6). We have also added a note to the discussion section (p. 14) to acknowledge the reviewer's point that the high-order dynamics we observe could reflect nonlinearities, chaotic patterns, nonstationarities or multi-stability,

etc. Importantly, whatever the “reason” for the presence of informative high-order dynamic patterns, our across-subject decoding results (e.g., Fig. 4, S1, S2) indicate that they are consistent across individuals. This suggests to us that they are stimulus-driven (e.g., following the logic of Simony et al. 2016) since non-stimulus-driven dynamics would be unlikely to exhibit correlated stimulus-locked spatiotemporal dynamics across people.

(3) On the same point, since these synthetic data contain only dynamic first order correlations, the behaviour of the higher order vectors X should be near null and this should be explored.

With respect to the simulations reported in Fig. 2, as the reviewer is aware, all multivariate timeseries data exhibit high-order patterns. Further, if low-order correlation dynamics are temporally structured in a reliably similar way across individuals, high-order correlation dynamics will also be temporally structured. The challenge is that if defining those high-order dynamics is not part of the generative process that produces the data, we have no way of comparing the recovered high-order dynamics to the “true” high-order dynamics. Therefore, as mentioned above, we have added simulations that exhibit known second-order correlation dynamics (Fig. 3) to help us explore the performance of our approach.

We also note that the key question of interest (in our explorations of BOLD data) is whether those high-order correlations contain additional structure beyond that reflected in first-order correlation dynamics alone. Our explorations suggest that high-order dynamics are informative to across-subject classifiers (e.g., Figs. 4, 5, S1, and S2).

In addition, testing the algorithm on a block-design task fMRI data set could be informative in recovering known discrete change points.

This is an interesting suggestion. However, prior work (e.g., Baldassano et al., 2016, Neuron, along with other work from Uri Hasson’s group) suggests that the sorts of “naturalistic” tasks such as listening to story narratives or viewing movies that appear to reliably evoke structured high-order dynamics exhibit “event boundaries” at different moments, depending on which brain regions is being considered.

We’re aware of ongoing work from other groups where participants view a series of short videos; that sort of dataset could be used in principle to test our

approach's ability to detect known change points (e.g., between each successive video). However, other than exploring synthetic data with known changepoints (Figs. 2, 3), we feel that collecting a new naturalistic dataset with known changepoints is beyond the scope of our current manuscript. Nevertheless, this could be an interesting focus of future work, and we have added a note to this effect in our discussion (p. 14).

2. It is not clear what the analyses with the “normalized accuracy” (Fig 5 RHS) show – it seems here that the resting state analyses can be decoded quite well with low order correlations – but why normalize to have unit maximum? Doesn't this artificially inflate those results? Why not use a statistical approach such as permutation or bagging?

The normalized accuracy plots factor out differences in absolute decoding accuracy across experimental conditions. Our intention is to make it easier for readers to visualize differences in relative decoding accuracy across features at different orders. We display the un-normalized (Fig. 4A, C) and normalized (Fig. 4B, D) accuracies next to each other so that readers can visually integrate them. The un-normalized accuracies highlight differences in decoding accuracy across experimental conditions, whereas the normalized accuracies highlight differences in decoding accuracy across different orders of features.

3. Related to these preceding concerns, the presence of non-stationarities in low order correlations will likely propagate to apparent (spurious) changes in higher order correlations. Therefore, I suggest the authors use a resampling scheme in the Fourier [2] or wavelet [3] domain and benchmark their empirical findings against these null (which preserve all trivial auto-and cross-correlations but destroy higher order ones). This may be particularly helpful in explaining the presence of low order inter-subject correlations in the resting state data (Fig. 6, right hand column).

The underlying question here is whether the high-order correlation dynamics we report are “real” properties of brain data that reflect meaningful underpinnings of cognition, or whether they are solely artifacts of nonstationarities or non-linearities in the low-order data.

Our revised simulations (Fig. 3) show that our approach *can* recover high-order dynamics if they exist in the data, although this does not specifically show whether or how those high-order dynamics are related to cognition. Conversely, our analyses of BOLD data cannot show whether we were able to accurately recover “true” high-order dynamics in the data (since the ground truth high-order

correlations are unknown). But we can use the BOLD data to identify stimulus-driven high-order brain patterns that are reliably similar across participants, and examine how the stability of those patterns change across experimental conditions that vary the temporal structure of the stimulus. We show that when the stimulus is richly structured in a meaningful way (intact and paragraph-scrambled conditions, Figs. 4, S1, S2) high-order brain patterns are sufficiently similar across people that they enable us to decode (across groups of participants) when in the stimulus the participants are listening to. During less richly structured stimulus conditions (word-scrambled and rest conditions), high-order patterns become less reliably similar across people and therefore are less informative to neural decoders.

The question of whether the stable high-order dynamics we identify during the intact and paragraph-scrambled conditions arise from physiological properties of the brain, or whether they solely reflect dynamic properties of the stimulus is interesting (although it is beyond the scope of our manuscript). We can say that not *all* BOLD data display stable high-order dynamics (e.g., word-scrambled and rest conditions), which we believe gets at a similar point to what the reviewer is proposing re: scrambled data. However, neither our analyses nor (to our reading and understanding) the ones the reviewer is proposing can identify the *cause* of those high-order dynamics. We have added a note to this effect in our discussion (p. 14).

One result that suggests that the high-order dynamics we observe are not an artifact is summarized in Figs. 6 and 7. In particular, the specific brain regions that most strongly drive high-order dynamics change systematically across the different experimental conditions. Whereas listening to a word-scrambled story (i.e., a stimulus that carries meaningful structure only over short timescales) leads to stable neural dynamics only in low-level auditory areas, more coarse scramblings (paragraph-scrambled and intact conditions; i.e., that have meaningful structure over longer timescales) lead to stable high-order neural dynamics in visual areas and frontal (executive function, high-order cognition, and cognitive control) regions. If the high-order dynamics we observed were an artifact of the stimulus rather than a true reflection of neural processing, we would not expect to see interpretable and systematically varying anatomical differences across the different experimental conditions.

Minor;

Abstract:

1. *Isn't the first sentence of the abstract a bit tautological? Our thoughts arise from neural things that change with our thoughts ...??*

We are simply referring to the (we think widely accepted) notion that thoughts arise from neural dynamics that reflect our ongoing experiences.

2. *The "brain's connectome" refers to the structural connections of the brain [6] – not the changes in functional connectivity referred to here*

We have changed this sentence to "...subgraphs of the brain's *functional* connectome" (p. 1, emphasis added).

Introduction:

1. *The notion that cognition is mediated by dynamic interactions between brain structures is much older than these cited references suggest – e.g. [4,5] but many others].*

Thank you for these suggestions; we have added citations of these papers (p. 2).

Methods:

1. *Line 91: Don't you mean "(T by $((K^2-K)/2+K)$ -dimensional)"?*

That line is referring to a single timepoint's reshaped correlation matrix (now p. 15).

2. *The discussion of dimensionality reduction on lines 127-148 is interesting but it's not strictly required for the paper and could be deleted. Likewise, for the discussion of graph metrics, lines 158-175.]*

The distinction between our two approaches for estimating high-order dynamic correlations (dimensionality reduction vs. graph metrics) is critical for making sense of Fig. 6 (which specifically requires the graph metrics approach). Our hope is that the discussion we included will help some readers to better understand our approach.

Discussion:

1. *Is figure 8 schematic only? That is not evident from the text or caption.*

We have clarified in the figure caption that our former Figure 8 (now Fig. 7) is a schematic.

References:

1: *Heitmann, S., & Breakspear, M. (2018). Putting the “dynamic” back into dynamic functional connectivity. Network Neuroscience, 2(02), 150-174.*

2: *Prichard D, Theiler J (1994) Generating surrogate data for time series with several simultaneously measured variables. Physical Review Letters, 73:951-954.*

3: *Breakspear, M., Brammer, M. J., Bullmore, E. T., Das, P., & Williams, L. M. (2004). Spatiotemporal wavelet resampling for functional neuroimaging data. Human brain mapping, 23(1), 1-25.*

4: *Friston, K. J. (2000a) The labile brain. I. Neuronal transients and nonlinear coupling. Phil. Trans. Roy. Soc. Lon., 355B, 215–236.*

5: *Grossberg, S. (1988). Nonlinear neural networks: Principles, mechanisms, and architectures. Neural networks, 1(1), 17-61.*

6: *Sporns: Sporns, O., Tononi, G., & Kötter, R. (2005). The human connectome: a structural description of the human brain. PLoS computational biology, 1(4), e42.*

Reviewer #2 (Remarks to the Author):

In this manuscript, Owen et al. introduce and test a set of methods for analyzing neural dynamics across multiple “orders” of co-variation within and across brains. In particular, they examine “high order dynamic correlations”, which are brain subnetworks which exhibit correlated changes across individuals. They apply their new method to brain dynamics from resting conditions, and also from task data in which participants listened to a narrative stimulus scrambled at the scales of words, sentences and paragraphs. Overall, they conclude that the higher-order correlation patterns contain more useful information for decoding the timepoints of the more

“cognitive rich” stimulus states, whereas the lower-order correlation patterns are ideal for decoding brain states associated with more basic, stimulus driven representations.

Overall, the concept is extremely intriguing and the paper is fairly well-written. At the same time, more work will be needed, both in simulation and also in the empirical analyses, in order to substantiate the central claims, and clarify the logic of some of the analyses.

MAIN POINTS

[1] The simulations used to validate the methodology are only used to test different kernel widths; however, the most innovative aspect of the paper is the recursive “higher order” correlation method, illustrated in Figure 1. The authors should conduct simulations to test (i) whether the higher-order correlation methods are indeed picking up on the structures (e.g. graph-state covariations) that are described visually in Figure 1, and (ii) that this translates into “optimal” temporal decoding of higher-order covariation structure information from higher order correlations. Such simulations could provide a basic validity support for the key logic that is applied to the empirical data in Figures 4-6.

This is a great suggestion that was also made by Reviewer 1. We have added new simulations of high-order dynamic correlations (Fig. 3) that highlight our ability to recover known high-order structure in synthetic datasets with different temporal properties (analogous to Fig. 2).

[2] In the original empirical analysis by Simony et al, the number of “reliably responding” voxels and “reliably inter-subject correlated” voxels for these data was shown to increase across conditions, with “word scramble” producing the smallest number of reliable voxels, and “intact” providing the largest. Could it be the case that the “higher order correlation” method does better when there is a larger number of inter-subject reliable voxels? Could the total number of reliable voxels per condition be confounded with the “optimal order” for decoding at that level? It seems that this issue could be directly resolved using simulation.

The reviewer is making a good point that when low-order dynamics are unreliable or unstable across people, we cannot hope to observe stable high-order dynamics across people (since high-order dynamics are estimated from

low-order dynamics). One way this can be seen is in our new simulations (Fig. 3)-- our ability to recover second-order patterns is always strictly worse than our ability to recover first-order patterns from the same moment in time. So the reviewer is correct that high-order dynamics will not provide useful information to neural decoders if the corresponding low-order dynamics are already unstable. The question of interest for us in this manuscript is at which "order" the most reliable and stable patterns are, under different experimental conditions. In other words, for each experimental condition, we want to know which orders of neural dynamics are reliable and stable across people.

[3] In Figure 5, it appears that "Rest" has a "normalized accuracy" score greater than 80%, and this is a little jarring. The text does not specify how the accuracy normalization is performed. Is it divisive? Please specify precisely. It is difficult to understand what one is supposed to take away from Figures 5B and 5D. The error ribbons appear to overlap for all curves except "Intact". In fact, the "rest" decoding appears to numerically exceed most conditions. Related to this, the text

Reviewer 1 made a similar point; we're copying and pasting our response here for convenience:

The normalized accuracy plots factor out differences in absolute decoding accuracy across experimental conditions. Our intention is to make it easier for readers to visualize differences in relative decoding accuracy across features at different orders. We display the un-normalized (Fig. 4A, C) and normalized (Fig. 4B, D) accuracies next to each other so that readers can visually integrate them. The un-normalized accuracies highlight differences in decoding accuracy across experimental conditions, whereas the normalized accuracies highlight differences in decoding accuracy across different orders of features.

[4] The primary results are generated by averaging across kernel widths. I understand that the authors do not wish to share all of the complexity that comes with high-dimensional analyses..... but at the same time, the averaging across kernel widths leaves the reader with the feeling that the main results vary substantially across kernel widths. Is it possible to report for which specific kernel widths the primary results (intact condition better decoded by higher order correlations) will hold?

Our goal in averaging across kernel widths and shapes was to identify results that were robust to the (arbitrary) analysis decisions about kernel parameters.

We have added additional supplementary figures that display how our results vary with kernel shape (Fig. S1) and kernel width (Fig. S2).

[5] Suppose that the decoding target was relaxed in time so that, instead of decoding a specific TR, the task was instead to decode (say) a specific 10-15 second window. Might this resolve some of the variability in the results related to different kernels? In particular, it may be more difficult for very temporally coarse kernels to succeed in decoding temporally precise targets? But if we make the target of decoding less temporally specific, then perhaps a wider range of kernels could produce results that are most consistent with one another?

There do seem to be some complex nonlinear interactions between kernel parameters and our decoding results (e.g., see Figs. S1, S2). We think this arises from several factors. As the reviewer notes, temporally precise decoding requires temporally specific (narrow) kernels. However, narrower kernels are also more susceptible to time-varying noise. Therefore wider kernels tend to lead to more stable (but less temporally precise) estimates. We achieve peak decoding performance when we use kernels somewhere in the middle of the range we explored (Fig. S2).

The reviewer's suggestion to use a 10--15 second window is interesting as well. We think it would work well for the "intact" and "paragraph-scrambled" conditions, for which the stimulus appears to exhibit meaningful structure for timescales on the order of 10s of seconds. However, for the "word-scrambled" condition, the stimulus scrambling occurs at the sub-TR level (words tend to persist for on the order of 0.5--1s, and the scan sequence used a 1.5s TR interval). Loosening our decoding criterion would therefore unfairly bias our results in the word-scrambled condition (relative to the paragraph-scrambled and intact conditions).

[6] line 367: "This dataset provided a convenient means of testing our hypothesis that different levels of cognitive processing and engagement are supported by different orders of brain activity dynamics." — Please reconsider the use of the phrase "supported by" here. Since the data are analyzed using a decoding analysis, it is unclear whether we could infer whether particular brain dynamics are causal in supporting particular cognitive states. If the authors want to make this causal claim, they should buttress the causal claim with argument.

This is a fair point. We have changed “supported by” to “reflected in” throughout our revised manuscript.

[7] I must confess that I had great difficulty understanding what is shown in Figure 7, to the extent that I cannot really comment on the results at all. The methods state: “We constructed a map of RBF components that denoted the endpoints of the 10 strongest correlations (we set each RBF to have a maximum value of 1).” But I cannot understand what is meant by an “endpoint” for correlations of (say) order 3. Should we expect this endpoint to correspond simply to a single region? Single regions are indeed plotted in the left-hand side of Figure 7. But, I had thought that the higher order correlations no longer involved single regions? Moreover, the methods that: “ However, attempting to map a given X_n back onto the original feature space of X_0 will usually require $O(K^2n)$ space and therefore becomes intractable as n or K grow large.” If it is not possible to map back from the higher order correlations to the original feature space, then I am a bit confused as to how the zero-order and higher-order correlation “endpoints” can be plotted on the same brain surface in Figure 7. Can the “endpoint” of a 4th-order correlation still be a single brain region?

We thank the reviewer for raising this concern. The key insight needed to make sense of Figure 7 (now Fig. 6) is that we are using the “graph measure” approach to estimating high-order correlations for that figure (method described starting on p. 19). Whereas the dimensionality reduction approach loses the mapping back to the original feature space, our graph measure approach retains the features in the original space. The “endpoints” we refer to in that figure caption are pairs of RBF nodes whose high-order correlations with each other are stable across individuals. We have added some additional text to help clarify this point (p. 28; subsection entitled *Reverse inference*).

SMALL POINTS

— Figure 4 contains “gray” markings in the legend, for narrow kernels, but there does not appear to be any gray coloring at all in the figures; since it only appears to be the Gaussian kernel whose width is varied in the figures (?), why not provide varying saturations of green to indicate changing kernel widths?

The notation in the legend is intended to show that the shading (represented by grayscale squares in the legend) reflects the kernel widths, whereas the color (represented by colored squares in the legend) reflects the kernel shapes. We include the following description in the Fig. 2 (i.e., Fig. 4 in our previous

submission) caption: “Different colors denote different kernel shapes, and the shading within each color family denotes the kernel width parameter.”

— *Figure 4 should contain a “ground truth” curve, indicating when exactly the event transitions / ramps /decays occurred in the underlying ground-truth covariance matrices. This will enable a more direct visual comparison between the timing of the simulation and the recovered time-courses from the analysis.*

Added.

— *The colormap for the t-values in Figure 6 is confusing... in particular, small positive t-values are shown in pale blue, while negative t-values are shown in dark blue. Am I missing something? Is the white color supposed to correspond to $t = 0$? Surely it would make sense to use a colormap that is perceptually symmetric around zero?*

We have updated the color map to show positive values in red and negative values in blue (Fig. 5 in our revised manuscript).

— *What is “decoded” during the resting scan? Please clarify that, just as with the task-locked scans, the “timepoint” is what is decoded during rest. This would be consistent with the interpretation of decoding performance in terms of time-varying changes in arousal and engagement over the course of the rest period.*

We have clarified that we are displaying *timepoint* decoding accuracy (Fig. 3, S1, S2, also p. 4, etc.).

— *How does the dimensionality of the data input to the PCA vary across “orders” of the analysis? In other words, if one were to plot the % variance explained as a function of number of principal components, how does the shape of this curve vary as one iterates from lower-order to higher-order correlations? Do the dynamics become more “low-dimensional” / compressible at higher orders?*

We have added a new set of analyses to address these questions (Fig. S7, partially copied below). We found that *lower-order* dynamics were more compressible (lower dimensional) than higher-order dynamics. We also found differences between the dynamics we observed during the listening experimental conditions (intact, paragraph, and word) versus during the rest condition. Overall the dynamics during the listening conditions tended to be more compressible (across a range of orders) than the dynamics during the rest condition. In Fig. S7

(and the portion of the figure copied below), curves with steeper slopes denote

dynamic patterns that are more compressible:

— “which types of neural activity patterns might comprise the neural code” ... this might read better as “which types of neural activity patterns might compose the neural code”

(The code comprises patterns, but the patterns compose the code)

Fixed.

— there are some very long “methods review” paragraphs (e.g. lines 161-174) in which long lists of related methods are cited. Since this is not a review paper, it is not clear that such long lists, with little commentary, belong in the main text of an empirical article. Is it possible to trim these lists to discuss only the most relevant related work?

Reviewer 1 made a similar point; copying and pasting our response:

The distinction between our two approaches for estimating high-order dynamic correlations (dimensionality reduction vs. graph metrics) is critical for making sense of Fig. 6 (which specifically requires the graph metrics approach). Our hope is that the discussion we included will help some readers to better understand our approach.

Reviewer #3 (Remarks to the Author):

Owen et al. develop a novel method for estimating higher-order correlations in fMRI activity, and validate it using simulated data. They then characterize higher-order correlation patterns in fMRI data collected as people listened to intact and scrambled stories and rested, and found that classifiers performed better for predicting the intact story timepoint when they used higher-order correlation features. In contrast, classifiers performed better for predicting time for the scrambled and rest condition when they only included activity or first-order correlation patterns.

Although the paper represents a helpful methodological advance in that it will allow researchers to characterize higher-order correlations quickly and efficiently, the robustness of the classification results and theoretical advance is less clear. My main comments are described in detail below.

Major comments:

1. The primary conclusion of the paper is that “high-level cognition is supported by high-order brain network dynamics.” “High-level cognition”, however, is never defined, and sometimes the intact story is said to reflect “cognitive engagement”, again a vague term. Is the claim that higher-order correlations reflect and/or support cognitive processes related to story understanding, like attention, learning, and memory, but not lower-order processes like perception? Clearly defining the process or processes being reflected here would increase the impact of the work.

We have clarified the distinction between “low-level cognition” versus “high-level cognition” on page 4 of our revised Introduction. These distinctions, and our rationale, are drawn directly from Uri Hasson’s work on temporal receptive windows using stimuli that are scrambled at different timescales (e.g., Hasson et al., 2008, Journal of Neuroscience):

“Temporal scrambling has been used in a growing number of studies, largely by Uri Hasson’s group, to identify brain regions that are sensitive to higher-order and longer-timescale information (e.g., cross-sensory integration, rich narrative meaning, complex situations, etc.) versus regions that are primarily sensitive to low-order (e.g., sensory) information. For example, Hasson et al. (2008) argues that when brain areas are sensitive to fine versus coarse temporal scrambling, this indicates that they are “higher order” in the sense that they process contextual information pertaining to further- away timescales. By contrast, low-level regions, such as primary sensory cortices, do not

meaningfully change their responses (after correcting for presentation order) even when the stimulus is scrambled at fine timescales.”

Extending these ideas to our work, we are using analogous logic about the meaning of responses to temporally scrambled stimuli to identify patterns of network dynamics that are modulated in response to high-order cognitive structure (e.g., versus solely in response to low-level sensory information).

2. Related to the first point about definitions, I think a longer discussion in the introduction about high-order correlations would be helpful. Is there an intuitive way to understand correlations beyond 2nd order? Is there any theoretical reason to look at correlations between order 0 and 10 but not beyond? Although I recognize the authors' sophisticated and rigorous methodological work I am still struggling to understand what these high-order correlations represent.

We agree that the meaning of n th order correlations can be difficult to conceptualize. The general logic is that n th order correlations reflect correlations between order $n - 1$ features (where order 0 represents the original data). In this way, “orders” are defined iteratively. We used the notion of homologous networks to illustrate this idea in our introduction (p. 2--3):

“Second-order correlations reflect *homologous* patterns of correlation. In other words, if the dynamic patterns of correlations between two regions, A and B , are similar to those between two other regions, C and D , this would be reflected in the second-order correlations between $(A-B)$ and $(C-D)$. In this way, second-order correlations identify similarities and differences between subgraphs of the brain's connectome. Analogously, third-order correlations reflect homologies between second-order correlations— i.e., homologous patterns of homologous interactions between brain regions. More generally, higher-order correlations reflect homologies between patterns of lower-order correlations.”

One reason *why* one might expect to see homologous networks in a dataset is related to the idea that network dynamics reflect ongoing neural computations and processing (e.g., the flow of information through our brain's networks; p. 3):

“One reason one might expect to see homologous networks in a dataset is related to the notion that network dynamics reflect ongoing neural computations or cognitive processing (e.g., Beaty et al., 2016). If the nodes in two brain networks are interacting (within each network) in similar ways then, according to our characterization of network dynamics, we refer to the similarities between those patterns of interaction as higher-order correlations. When higher-order correlations are themselves changing over

time, we can also attempt to capture and characterize those high-order dynamics.”

There is no theoretical reason to only examine correlations between orders 0 and 10 (in fact, we examine correlations up to order 15 in Figs. S3--S6). Our decision to examine patterns up to order 10 in the main text was motivated by our finding that our classifiers seemed to be topping out around order 4, and we wanted to examine at least some patterns at orders above 4 to rule out potential (local) non-monotonicities.

3. The paper’s main conclusion relies on qualitative comparison of decoding accuracy models with higher-order correlation features for the story condition but lower-order correlation features for the rest condition. However, it is hard to interpret how meaningful these differences are, especially since Figure 5 (left panel) appears to show stable accuracy regardless of the feature orders included in the model for all scan conditions. Are differences in model accuracy as a function of correlation orders included statistically significant? Are they consistent across additional splits of the data? (Currently there are only 10, and based on Figure 5 there appears to be high variability across splits.) If not, the broad conclusion that higher-order feature models are better for the story condition but lower-order features are better for scrambled stories and rest may not be justified.

We agree with the reviewer that the differences in decoding accuracies between classifiers trained on different orders are relatively small in magnitude. We tested our hypothesis in several ways:

- **In Figures 4 and 5 in the main text we carry out quantitative (statistical) comparisons of decoding accuracies of models that include different orders of features. Relative to the average decoding accuracy, there is little variation across cross-validation folds (e.g. compare the sizes of the error ribbons in Figs. 4a and 4c to the overall heights of the curves). We have also added additional supplementary figures (Figs. S1 and S2) in order to break down these results further by kernel shape (Fig. S1) and kernel width (Fig. S2). In each of these figures, we average the results over all cross validation folds and model parameters (i.e., kernel shapes and widths), unless otherwise specified (as in Figs. S1 and S2). This enables us to normalize over potentially idiosyncratic results that arise from a particular kernel shape or width, and instead focus on results that are robust to those analytic choices. Figure 4b and 4d “zooms in” on each curve (normalizing for overall decoding accuracy so that all of the curves lie on top of one another) to make their individual shapes easier to**

visualize. Whereas Figures 4a and c emphasize differences in decoding accuracy across *conditions*, Figures 4b and d emphasize differences in decoding accuracy across *orders*.

- The diagonals in Figure 5 indicate the average feature importance (weight) given to features of each order, in each experimental condition (darker green = more importance; lighter shading = less importance). We can see that the green shading extends further down the diagonal when decoding the “intact” conditions versus decoding the scrambled (“paragraph” and “word”) conditions. In other words, higher-order features are more informative for decoding listening times from the intact story than from the scrambled stories.
- The specific brain structures that mediate network dynamics at different orders change in an interpretable way across the different experimental conditions (Fig. 6; summary cartoon in Fig. 7; Figs. S3--S6 provide additional detail).

Taken together, our analyses show that although the absolute differences in decoding accuracy are modest, (a) our main results are robust to a range of model parameter choices and across different cross-validation folds, (b) our main results can be seen both in decoding accuracy (Fig. 4) and feature weights (Fig. 5), and (c) the specific brain regions we identify as mediating the dynamic patterns at different orders vary in a systematic and intuitive way across experimental conditions.

4. The article argues that high-order correlations contribute to predictive power during an intact story condition but not at rest. However, the bottom right matrix in Figure 6 perform less well for classifying timepoint than higher-order correlations (up to order 5). This appears counter to the argument that lower-order features are better for classifying time point during rest. Is this interpretation warranted?

There is some nuance to our resting state decoding results that we have clarified in our revised manuscript. Our analysis framework has two basic pipelines-- in the “PCA pipeline” (e.g., Figs. 4a, b; Fig. 5 top row), we examine dynamic high order correlations in *activity* patterns. When characterized in this way, neither dynamic first-order correlations nor dynamic higher-order correlations are informative to resting-state temporal classifiers (Figs. 4a and 4b, purple lines; Fig. 5, top right matrix).

In the “Eigenvector centrality pipeline” (e.g., Figs. 4c, d; Fig. 5 bottom row), we examined high order correlations in *centrality* patterns. In other words, the eigenvector centrality analyses examined dynamic correlations in different nodes’ *positions* in the broader network (as measured by each node’s timeseries of eigenvector centrality values at different orders). When characterized in this way, we still find that the most informative features’ orders are higher for the intact story listening condition than for the rest condition (e.g. compare pink and purple lines in Fig. 4c; bottom left vs. bottom right matrices in Fig. 5). Specifically, fourth-order dynamic correlations are most informative to temporal classifiers during the intact condition, but second-order dynamic correlations are most informative to temporal classifiers during the rest condition. Although second-order correlations are still “high-order correlations,” we have clarified that our note refers to the *relative* orders, not the *absolute* orders (e.g., p. 4).

5. Given that head motion can be a significant confound in functional connectivity (and especially dynamic functional connectivity studies), are high-order correlation patterns affected by data quality or quantity (e.g., number of frames available, head motion)? Could these differences explain the differential contribution of different correlation orders to decoding accuracy across scan conditions? For example, could subjects have moved their heads more—or more consistently—during rest than stories, or vice versa? Are estimates of high-order correlations systematically affected by time series length, and did scan lengths differ across conditions?

The reviewer brings up several important points here. The most critical concern we identify in the reviewer’s comment is that the decoding results (using different orders of dynamic correlations in different experimental conditions) might be susceptible to non-neuronal artifacts, such as head motion, heart rate or breathing rate variability, etc. Although we agree that these measures can influence the patterns we are measuring *within* individual participants’ brains, the patterns we are feeding to our temporal decoders are computed *across* individuals in a way that filters out patterns that are uncorrelated across people (including motion, heart rate variability, breathing rate variability, etc.). Specifically, our dynamic correlation measures are a generalization of the inter-subject functional connectivity approach proposed by Simony et al. (2016, Nature Communications). As those authors discuss (and elaborate on in Simony and Chang, 2020, NeuroImage), these inter-subject measures serve to highlight neural patterns that are specifically stimulus-driven, and filter out patterns that are not stimulus-driven. The logic in that prior work, which also extends to our work, is that stimulus-driven activity should be correlated across people, whereas

non stimulus-driven activity should not be correlated across people. Computing dynamic (lower-order and higher-order) correlations *across* people enables us to focus in specifically on these stimulus-driven patterns.

One aspect of our data the reviewer identifies that is *not* controlled for by computing correlations across (rather than within) participants is the number of frames (i.e., fMRI volumes) in each condition. Our simulations (analogous to Figs. 2 and 3) indicate that our ability to recover ground truth low-order and high-order dynamic correlations is not substantially affected by the number of samples, provided that the number of samples is much wider than the kernel width (also see Fig. S2). In our analyses of fMRI data, however, the number of functional volumes (frames) did differ somewhat across conditions: intact (300 TRs), paragraph (272 TRs), word (300 TRs), and rest (400 TRs). To address this difference, we correct for non-signal and non-feature related differences in the number of volumes when we normalize the reporting decoding accuracies by the estimated “chance” decoding accuracy in each condition (e.g., see Fig. 4 caption). We also observed no meaningful effects of the number of frames; e.g. our main finding, that listening to the richer (intact story) stimulus is reflected in higher-order dynamic correlations than listening to the more impoverished (scrambled) stimulus, hold when considering only the “intact” and “word” conditions, which had equal numbers of fMRI volumes.

*6. The paper’s methods are complex, and include defining network nodes with HTFA, estimating dynamic functional connectivity with different kernels, projecting these patterns in low-dimensional space using two approaches, repeating this process for up to 10th-order correlations, temporal decoding with the resulting matrices after a parameter search to optimize *n*th-order matrix weights, averaging model performance across 12 classifiers with different combinations of kernels and kernel widths. Especially in light of this complexity the figures are a good opportunity to walk readers through the analysis pipeline. However, the current figures are a bit hard to follow. For example, Figure 1 does a nice job explaining the difference between fMRI activity vs. functional connectivity vs. network-to-network interactions. However, it is difficult to extract the most relevant information for understanding the rest of the paper from this graph. Is it necessary to distinguish within- from across-brain relationships for the current work? Univariate vs. multivariate relationships? What exactly is meant by a “homology”?*

The distinction between within-brain versus across-brain patterns is important, because the across-brain aspect of the analyses we carry out on the fMRI data is

what enables us to specifically focus in on stimulus-driven patterns (we also note this in our response to the reviewer's previous comment).

The univariate versus multivariate distinction is also important. Univariate network patterns reflect one regions' "connectivity" (i.e., correlations and/or centrality) with respect to other nodes in the network. Multivariate patterns reflect the full set of univariate interactions (e.g., across all nodes in the network). We felt that, given the complexity of the patterns we are describing, it would be useful to build up incrementally from univariate to multivariate patterns.

Finally, homologies reflect sets of networks that exhibit similar connectivity dynamics. Reviewer 1 asked a similar question about the nature of homologies, and we have copied the relevant portions of our response to their comment below for convenience:

We agree that the meaning of n th order correlations can be difficult to conceptualize. The general logic is that n th order correlations reflect correlations between order $n - 1$ features (where order 0 represents the original data). In this way, "orders" are defined iteratively. We used the notion of homologous networks to illustrate this idea in our introduction (p. 2--3):

"Second-order correlations reflect *homologous* patterns of correlation. In other words, if the dynamic patterns of correlations between two regions, A and B , are similar to those between two other regions, C and D , this would be reflected in the second-order correlations between $(A-B)$ and $(C-D)$. In this way, second-order correlations identify similarities and differences between subgraphs of the brain's connectome. Analogously, third-order correlations reflect homologies between second-order correlations— i.e., homologous patterns of homologous interactions between brain regions. More generally, higher-order correlations reflect homologies between patterns of lower-order correlations."

One reason *why* one might expect to see homologous networks in a dataset is related to the idea that network dynamics reflect ongoing neural computations and processing (e.g., the flow of information through our brain's networks; p. 3):

"One reason one might expect to see homologous networks in a dataset is related to the notion that network dynamics reflect ongoing neural computations or cognitive processing (e.g., Beaty et al., 2016). If the nodes in two brain networks are interacting (within each network) in similar ways then, according to our

characterization of network dynamics, we refer to the similarities between those patterns of interaction as higher-order correlations. When higher-order correlations are themselves changing over time, we can also attempt to capture and characterize those high-order dynamics.”

Similarly, the take home messages are not easy to extract from Figures 5 and 6. Figure 5 shows minimal differences between classifiers that include different order features, thus suggesting that decoding is relatively stable within a scan condition regardless of the features used, a point contrary to the main message of the paper.

Although we concede that the absolute magnitude of the differences in accuracy between classifiers that include information at different orders is relatively small, we showed that those differences are meaningful in three ways (we expand on these ideas in our response to the reviewer’s third comment above): relative decoding accuracy as a function of order, across experimental conditions (Fig. 4b, d); relative weighting given to different orders of dynamic correlations across different experimental conditions (Fig 5., diagonals in all of the matrices); and neurosynth-based meta analyses that show meaningful functional difference between the networks that mediate dynamic correlations at different orders, in different experimental conditions (Fig. 6, S3--S6).

We also note that each classifier we ran (e.g., in Fig. 4) contains features *up to and including* the indicated order. Therefore the fact that classifiers beyond a given order also perform well is not necessarily indicative of higher-order patterns also containing meaningful information. This is why our other analyses that consider features at each order individually (Fig. 5, 6, S3--S6) are also important for interpreting our full set of findings.

Likewise, it took me several close reads of the Figure 6 caption to understand the organization and color scale, and I’m still not sure I could replicate how the value in each cell was calculated. Perhaps a schematic would help here? Or a step-by-step explanation? Some summary take-aways could also help too. For example, does the consistent blue pattern in the top left matrix show that lower-order features are always worse than higher-order features, and does the all-red pattern in the second matrix in the bottom row show that lower-order features are always better than higher-order features? The weights on the diagonal are a bit confusing in light of the manuscript’s claims about the value of higher-order correlations for decoding scan time point, too. Are results associated with correlations of orders greater than 4 meaningful or important given that their weights in predictive models are, on average, 0?

We appreciate the reviewer's efforts to understand that figure. Following the reviewer's suggestions, we have cleaned up the figure caption (now Fig. 5) to help clarify what is being shown, and we have also corrected our color scale to match the caption (we apologize for additional confusion caused by the color scale in our previous submission). We also unpack the figure in more detail here for convenience.

In the revised figure, the upper row of matrices displays results for the "PCA" (i.e., activity-based) version of our analysis pipeline, and the bottom row of matrices displays results for the "Eigenvector centrality" (i.e., centrality-based) version of our analysis pipeline. Each column of matrices displays results for one experimental condition.

Within each matrix, the upper triangles compare decoding accuracy obtained using different sets of features. Cooler colors (shades of blue) denote that features of the order indicated by the given row yielded *lower* decoding accuracy than features of the order indicated by the given column. Warmer colors (shades of red) denote that features of the order indicated by the given row yielded *higher* decoding accuracy than features of the order indicated by the given column. For example, when a row is colored entirely in blue (to the right of the diagonal), that indicates that the given row's feature is *less informative* than all of the other features we examined. Analogously, when a row is colored entirely in red (again, to the right of the diagonal), that indicates that the given row's feature is *more informative* than all of the other features we examined. When a given row contains a mix of cool and warm colors, that indicates that some features were more informative than that row's feature, and other features were less informative than that row's feature.

The lower triangles display the corresponding p -values. In other words, whereas the entries across the first *row* denote comparisons (t -tests) between order 0 features and order 1--10 features, the entries down the first *column* denote the corresponding p -values for those t -tests. Taken together, the upper and lower triangles in each matrix show how *relatively informative* each type of feature is when considered in isolation and compared to other features (also in isolation).

Finally, the diagonals of each matrix show the average optimal mixture weights assigned to features at each order when we trained classifiers using *all* features (order 0--10). Features shaded in darker green were weighted more heavily by those weighted classifiers. Taken together, the diagonals tell us about feature

importance and cross-feature redundancies. Because our estimation procedure produces noisier (less accurate, lower signal-to-noise) estimates of high-order features than low-order features, the classifier weight optimization procedure will weight lower-order features more heavily if they carry similar information as higher-order features (but with higher SNR). When high-order features are weighted more heavily, it indicates that those features carry *new* information beyond what is contained in lower-order features.

With the above explanation, we hope that it is now clear that the weights (green shaded squares) along the diagonal are conceptually different from the entries in the upper and lower diagonals of each matrix. Whereas the diagonals reflect weights obtained by training classifiers using *combinations* of features of different orders, the upper and lower triangles of each matrix display comparisons of results obtained using *individual* orders of features. Therefore the observation that features of a given order tend to receive 0 weight in the “combined feature order” classifiers is not strictly redundant with the “single feature order” classifiers, and the comparisons between those individual features can still be informative.

Minor comments:

1. Did the hierarchical topographic factor analysis (HTFA) approach used for node definition compromise the independence between training and test data? If all subjects were used to define the regions of interest then the test data would have influenced the features used at training. A completely independent approach would define the nodes in the training set and apply them to the test set, or use a pre-defined functional brain atlas developed in an independent sample.

HTFA must be fit jointly on *all* of the data in order to find a common set of nodes. Otherwise we would have no formal way of “matching up” the node labels in one participant’s data with other participants’ node labels. However, the per-timepoint node activations (i.e., the inputs to the analyses reported here) are estimated independently for each individual participant’s data. In this way, the HTFA step of the analysis does not compromise the independence between training and test data, just as utilizing a common set of ROIs, spatially warping individual participants’ brains to match a common template, and/or using a common kernel to spatially blur the data, would not compromise this independence.

2. Related to HTFA, Manning et al. (2018) previous shown that dynamic functional connectivity patterns examined with this approach can predict time points of an intact and scrambled story using the same data reported here. It would be helpful to acknowledge this prior work and clarify what the current work adds.

We have added several additional citations of Manning et al. (2018) in order to clarify which aspects of our analyses drew direct inspiration from that work. In addition, we include a paragraph in our discussion section explaining how our work extends the Manning et al. (2018) paper and other related work (p. 13--14):

“The notion that cognition is reflected in (and possibly mediated by) patterns of first-order network dynamics has been suggested by or proposed in myriad empirical studies and reviews (e.g., Chang & Glover, 2010; Demertzi et al., 2019; Fong et al., 2019; Gonzalez-Castillo et al., 2019; Lie ´geois et al., 2019; Lurie et al., 2018; Manning et al., 2018; Park et al., 2018; Preti et al., 2017; Roy et al., 2019; Turk-Browne, 2013; Zou et al., 2019). Our study extends this line of work by finding cognitively relevant *higher-order* network dynamics that reflect ongoing cognition. Our findings complement other work that uses graph theory and topology to characterize how brain networks reconfigure during cognition (e.g., Bassett et al., 2006; Betzel et al., 2019; McIntosh & Jirsa, 2019; Reimann et al., 2017; Sizemore et al., 2018; Toker & Sommer, 2019; Zheng et al., 2019).”

3. It is worth highlighting in the text that, in contrast to the majority of applications of machine learning approaches to fMRI, the decoding results do not provide predictions for individual subjects, but instead predict time points based on group-averaged data.

We have clarified the text throughout our manuscript where appropriate. For example (p. 4, emphasis added):

“We used a subset of the story listening and rest data to train across-participant classifiers to decode listening times (*of groups of participants*) using a blend of neural features (comprising neural activity patterns, as well as different orders of dynamic correlations between those patterns that were inferred using our computational framework).”

We have also added clarifying text to our revised methods section (e.g., sub-section entitled *Forward inference and decoding accuracy*, p. 24; also see Fig. 10).

4. The concluding remarks section is a bit overstated. (It reads: “The complex hierarchy of dynamic interactions that underlie our thoughts is perhaps the greatest mystery in modern science. Methods for characterizing the dynamics of high-order correlations in

neural data provide a window into the neural basis of cognition. By showing that high-level cognition is reflected in high-order network dynamics, we have elucidated the next step on the path towards understanding the neural basis of cognition.”) Although understanding the neural bases of cognition is certainly an important goal of science, the broad claims here almost undermine the authors’ interesting work by “protesting too much”.

We have substantially toned down the concluding remarks section as suggested (p. 15). Specifically, we retained the last sentence but removed the section header and the rest of that section.

REVIEWER COMMENTS

Reviewer #1 (Remarks to the Author):

I am satisfied with the authors' responses to my comments

Reviewer #2 (Remarks to the Author):

I thank the authors for their substantive revisions to the manuscript; I have just a few points remaining.

[1] Thank you for adding the new simulations (Figure 3), which help greatly to concretize the main claims and illustrate the strengths and weaknesses of the analytic approach. I do have two remaining questions about these simulation results:

(a): what is the limiting factor in achieving higher correlations between the ground truth values in the simulation and the practically recovered values? Is it just the length of the data samples? If this is the case, it could be helpful to make this clear to the reader, and it could be illustrative to show how the match with the ground truth approaches ceiling values as sources of noise are decreased and sample size is increased. [I recognize that this may render the synthetic data less realistic, but it seems important to demonstrate that the true underlying values can be recovered in the limit of infinite noiseless data.] On the other hand, if I am misunderstanding the reason for the (seemingly) low correlation between ground-truth and recovered values (e.g. please explain the source of this discrepancy), or the reason why the "ceiling" on these correlations is low.

(b) What is the cause of the "dip" at the start and "ramp" at the end of the Order-2 curves in Figure 3—"Constant" and Figure 3—"Ramping"? Are these finite-sample effects arising from the change in size of a sliding window / kernel at the start and end of the synthetic data? If so, please make this clear to the reader. Also, if these effects are present, might it make sense to "trim" the Order-2 empirical results to remove these finite-sample edge effects?

[2] In my original review I asked two related questions [these were both part of Question 2 from my original review]:

(2a) Could it be the case that the "higher order correlation" method does better when there is a larger number of inter-subject reliable voxels? And

(2b) Could the total number of reliable voxels per condition be confounded with the "optimal order" for decoding at that level?

In their response, the authors (as far as I can tell) answered (2a) but did not provide an answer to (2b). As they stated in their response: "For experimental condition, we want to know which orders of neural dynamics are reliable and stable across people." So my question in (2b) remains: whether the set of "which orders are reliable and stable" across conditions could be confounded (across conditions) with the total number of Order-1 reliable voxels. If the answer is "yes", I guess that's OK, but it seems that this potential confound should be made clear to the reader.

[3] The Introduction is still a little light on some of the (by now) classic work on network interactions and higher order cognition — the authors may wish to examine the two papers below (and related literature), to see whether the two papers usefully support or contextualize the central claim of this manuscript — that more complex inter-regional interaction patterns coincide with more complex cognitive function:

Coordination dynamics and cognition:

Bressler, S. L., & Kelso, J. S. (2001). Cortical coordination dynamics and cognition. *Trends in cognitive sciences*, 5(1), 26-36.

The concept of "neural context" [i.e. brain network context] for understanding high-level cognition:

McIntosh, A. R. (2000). Towards a network theory of cognition. *Neural Networks*, 13(8-9), 861-

870.

Reviewer #3 (Remarks to the Author):

I thank the authors for thoroughly addressing my comments. The manuscript is much clearer as a result of the changes made. I do continue to have some reservations about the strength of the claims drawn from relatively small differences in classifier accuracy between models including features of different correlation orders (in Fig. 4). However, the authors have done a good job demonstrating the validity and theoretical contribution of their new approach, and I think this manuscript will make an important contribution to the literature.

Reviewer #1 (Remarks to the Author):

I am satisfied with the authors' responses to my comments

(Nothing to address)

Reviewer #2 (Remarks to the Author):

I thank the authors for their substantive revisions to the manuscript; I have just a few points remaining.

[1] Thank you for adding the new simulations (Figure 3), which help greatly to concretize the main claims and illustrate the strengths and weaknesses of the analytic approach. I do have two remaining questions about these simulation results:

(a): what is the limiting factor in achieving higher correlations between the ground truth values in the simulation and the practically recovered values? Is it just the length of the data samples? If this is the case, it could be helpful to make this clear to the reader, and it could be illustrative to show how the match with the ground truth approaches ceiling values as sources of noise are decreased and sample size is increased. [I recognize that this may render the synthetic data less realistic, but it seems important to demonstrate that the true underlying values can be recovered in the limit of infinite noiseless data.] On the other hand, if I am misunderstanding the reason for the (seemingly) low correlation between ground-truth and recovered values (e.g. please explain the source of this discrepancy), or the reason why the "ceiling" on these correlations is low.

There are (at least) three main factors that negatively affect the correlations between the ground truth and recovered dynamic correlations (we have added a note to this effect to page 7 of our revised manuscript, along with two additional supplemental analyses summarized in Figs. S3 and S4):

- A. First, the procedure that is used to generate the correlations is itself noisy (based on random draws), so in practice what we refer to as the "ground truth" reflects the expected values from the data generation process, rather than the sampled values. Our characterization of the "ground truth" at each order becomes less accurate as the order decreases. This follows from how we generate synthetic data. The highest-order timeseries is generated first (via a noisy process), and each successively lower-order timeseries is generated via a noisy process from the next-higher-order timeseries. In this way, noise in our synthetic data *generation* procedure propagates from higher orders to lower orders. We think**

this is the primary reason that the recovered second-order correlations appear to be lower than the recovered first-order correlations.

- B. Second, our procedure for recovering dynamic (low-order and high-order) correlations is inexact. Because each order (n) of correlations is estimated using the next-lower order ($n - 1$), noise in our estimation procedure is compounded with each new order. In this way, *estimation* errors propagate from lower orders to higher orders. This means that the best possible recovery performance we can achieve at a given order is upper-bounded by the reconstruction accuracy of the next-lowest order.
- C. Finally, as the reviewer notes, our simulations are data-limited (e.g., we cannot generate timeseries data embedded with high-order correlations for very large values of T or K). The main challenge we face is that our procedure for generating timeseries data embedded with high-order correlations is very computationally expensive. The approach we used in our previous submission required storing $O(T * K^{2^{n+1}})$ values (where T is the number of timepoints, K is the number of features, and n is the order). After tweaking our synthetic data generation procedure (revised methods, pages 23--24), we were able to reduce the memory footprint to $O(T * K^{2^n})$. While our revised procedure is substantially more efficient (while still achieving similar performance), it is still not scalable to very large values of K or n . For example, generating second-order timeseries data of the same size as our (reduced) fMRI data from a single participant would require nearly 300 TB of memory, which is well beyond the capacity of the computing resources available to our lab. This limits our ability to directly test our ability to recover ground truth patterns in datasets of a similar size to the fMRI dataset we examine in the main part of the manuscript.

In addition to adding the above discussion points, we have added two supplemental analyses (Figs. S3 and S4) to directly examine the impacts of the number of samples (T) and the number of features (K) on our ability to recover ground truth first-order and second-order dynamic correlations from synthetic data. In summary, we found that as the number of samples increases, our ability to recover ground truth patterns improves (Fig. S3). This prediction is in line with the reviewer's prediction. Of note, when T is greater than between 250 and 500 samples (i.e., the range encompassed by our fMRI datasets), reconstruction accuracy starts to asymptote:

We also explored how reconstruction accuracy changes with the number of zero-order features (K). Up to the relatively small number of features we were able to test, our ability to recover ground truth patterns appears to be slightly impaired as the number of features increases (Fig. S4). We were somewhat surprised by this latter finding, since we had naively expected that recovery accuracy would *improve* with the number of features:

Exploring this issue more fully would require further optimizations and improvements to our synthetic data generation procedure. While this is beyond the scope of our current manuscript, it would be interesting to address in future work.

(b) What is the cause of the “dip” at the start and “ramp” at the end of the Order-2 curves in Figure 3—“Constant” and Figure 3—“Ramping”? Are these finite-sample effects arising from the change in size of a sliding window / kernel at the start and end of the synthetic data? If

so, please make this clear to the reader. Also, if these effects are present, might it make sense to “trim” the Order-2 empirical results to remove these finite-sample edge effects?

We have added a note to the manuscript (Fig. 3 caption) clarifying that the dips and ramps observed at the sharp transition points in our synthetic datasets reflect finite-sample “artifacts” as the reviewer notes. Nevertheless, due to the reasons outlined above (in response to the reviewer’s previous comment), our sense is that we are still under-estimating recovery performance, even at sharp transition points. If these finite-sample issues were simply adding “noise” to our reconstruction procedure, then we would have expected reconstruction accuracy to be worse at boundaries (whereas in practice we observe that reconstruction accuracy is *higher* near boundaries).

A second set of reasons for not trimming the recovered high-order timeseries estimates in the empirical data relate to what we are trying to learn from the fMRI data. We are specifically interested in how decoding accuracy varies when we consider different orders of features. To fairly compare across features, all of the features should be based on the same underlying data, contain the same numbers of samples, etc. Further, we would run into a practical limitation whereby trimming the beginning and end of the recovered timeseries at each successively increasing order would lead us to “run out of data” at higher orders.

In acknowledgement of noise in our estimation procedure, we have included a note (p. 8--9, emphasis added) that “[t]his suggests that our modeling approach provides a meaningful (*if noisy*) estimate of high-order correlations”.

[2] In my original review I asked two related questions [these were both part of Question 2 from my original review]:

(2a) Could it be the case that the “higher order correlation” method does better when there is a larger number of inter-subject reliable voxels? And (2b) Could the total number of reliable voxels per condition be confounded with the “optimal order” for decoding at that level? In their response, the authors (as far as I can tell) answered (2a) but did not provide an answer to (2b). As they stated in their response: “For experimental condition, we want to know which orders of neural dynamics are reliable and stable across people.” So my question in (2b) remains: whether the set of “which orders are reliable and stable” across conditions could be confounded (across conditions) with the total number of Order-1 reliable voxels. If the answer is “yes”, I guess that’s OK, but it seems that this potential confound should be made clear to the reader.

We have added a note to our discussion section to clarify this point (p. 15):

“One limitation of our approach relates to how noise propagates in our estimation procedure. Specifically, our procedure for estimating high-order dynamic correlations depends on estimates of lower-order dynamic correlations. This means that our measures of which higher-order patterns are reliable and stable across experimental conditions are partially confounded with the stability of lower-order patterns. Prior work suggests that the stability of what we refer to here as first-order dynamics likely varies across the experimental conditions we examined (Simony et al., 2016). Therefore a caveat to our claim that richer stimuli evoke more stable higher-order dynamics is that our approach assumes that those high-order dynamics reflect relations or interactions between lower-order features.”

[3] The Introduction is still a little light on some of the (by now) classic work on network interactions and higher order cognition – the authors may wish to examine the two papers below (and related literature), to see whether the se papers usefully support or contextualize the central claim of this manuscript – that more complex inter-regional interaction patterns coincide with more complex cognitive function:

Coordination dynamics and cognition:

Bressler, S. L., & Kelso, J. S. (2001). Cortical coordination dynamics and cognition. Trends in cognitive sciences, 5(1), 26-36.

The concept of “neural context” [i.e. brain network context] for understanding high-level cognition:

McIntosh, A. R. (2000). Towards a network theory of cognition. Neural Networks, 13(8-9), 861-870.

We appreciate the pointers to these papers and have added citations to both of them in our revised introduction (p. 2) and discussion (p. 14).

Reviewer #3 (Remarks to the Author):

I thank the authors for thoroughly addressing my comments. The manuscript is much clearer as a result of the changes made. I do continue to have some reservations about the strength of the claims drawn from relatively small differences in classifier accuracy between models including features of different correlation orders (in Fig. 4). However, the authors have done a good job demonstrating the validity and theoretical contribution of their new approach, and I think this manuscript will make an important contribution to the literature.

(Nothing to address)

REVIEWERS' COMMENTS

Reviewer #2 (Remarks to the Author):

I thank the authors for their thorough response. All of my concerns have been addressed.